# Efficiently Solving the TSP in One-Shot Without Post-Processing

## Abstract

Neural approaches relying on autoregressive (AR) solution construction for solving the Traveling Salesman Problem (TSP) — a cornerstone challenge in Combinatorial Optimization (CO) — can be computationally expensive during both training and inference, as each step of the solution construction requires a pass through the neural network. While non-autoregressive (NAR) methods promise efficient solution generation in just a single pass through the network, prior works have often required post-processing search techniques such as Monte Carlo Tree Search (MCTS), active search, beam search and 2-opt to achieve near-optimal performance. However, these techniques not only inflate inference times but have also drawn criticism for potentially overshadowing the model's intrinsic capability and raised questions about the contribution of the neural component in the solution generation pipeline.

To address these concerns, we demonstrate that high-quality solutions can be generated in one-shot using the NAR approach by producing highly accurate heatmaps without requiring sophisticated post-processing techniques. During inference, we decode the tour by selecting the top two edges per node based on the heatmap — a process which can be performed in parallel. Unlike most prior NAR approaches, which depend on costly expert solutions for supervised learning, our approach is trained with Self-Improvement Learning (SIL) which is entirely unsupervised. Trained as a unified model across graph sizes, our approach achieves near-optimal results. In addition, we introduce new metrics quantifying edge prediction precision and tour validity in the NAR setting, which emphasize the model's raw capabilities.

## 1 Introduction

The Travelling Salesman Problem (TSP) is an important problem in combinatorial optimization (CO), with widespread applications across domains such as logistics, robotics, genetics, and circuit design (Lenstra & Kan, 1975). As an NP-hard problem, TSP poses significant computational challenges. While exact solvers like Concorde (Applegate et al., 2006) can find optimal solutions for instances with tens of thousands of nodes, the computation time escalates dramatically with problem size. Heuristics-based methods such as (Helsgaun, 2017; Taillard & Helsgaun, 2019) have delivered impressive results — finding near-optimal solutions for problems with millions of nodes — but they rely heavily on hand-crafted rules and domain specific knowledge (Vesselinova et al., 2020).

In recent years, machine learning (ML) has gained traction as a promising paradigm for developing data-driven solvers that can address diverse CO problems without extensive manual engineering (Bengio et al., 2021). The TSP is conventionally formulated as a sequential decision-making process, where a tour is constructed incrementally by deciding the next city to visit at each step. Many recent learning-based methods adopt this approach by autoregressively constructing the tour guided by a neural network model (Vinyals et al., 2015; Bello et al., 2017; Kool et al., 2019; Kwon et al., 2020; Luo et al., 2023; Drakulic et al., 2023). However, this strategy can be slow and computationally expensive during both training and inference, as it requires querying the neural model at each step of the solution construction, thereby limiting scalability.

Non-autoregressive (NAR) alternatives have been proposed which mitigate the inefficiencies of the AR approach, where the neural network outputs probability distributions over the edges of each node

in a single pass. These probability matrices are commonly known as *heatmaps* in the literature (Joshi et al., 2019). However, these heatmaps have required post-processing techniques such as beam search, Monte Carlo Tree Search (MCTS) (Browne et al., 2012), or local optimization such as 2-opt to achieve high-quality solutions (Fu et al., 2021; Qiu et al., 2022; Min et al., 2023; Sun & Yang, 2023). These techniques not only increase inference times but also obscure the model's intrinsic capabilities, raising questions about the neural component's true contribution (Xia et al., 2024). Although NAR methods outperform AR counterparts in efficiency by generating the heatmaps in one-shot, they still rely on some sequential decoding (e.g., greedy decoding of the heatmap) to guarantee valid TSP tours (i.e., Hamiltonian cycles), even if powerful post-processing is avoided.

Apart from the inefficiencies of the AR approach, we also raise a fundamental question about the necessity of sequential construction for solving the TSP. Consider an autoregressive model capable of producing optimal TSP solutions step by step. For the entire predicted tour to be optimal, each step must be optimal, including the first one. Due to the cyclic nature of TSP, the optimal solution is invariant to the starting point. Consequently, the model must produce the optimal tour irrespective of the starting node. Therefore, the model must be able to predict the optimal first step from any node, as the optimal tour can begin from any node in the cycle. In theory, this implies that by predicting the optimal first step from every node in parallel — without relying on partial tour context — the entire tour could be reconstructed in a single pass, eliminating the need for autoregressive generation.

Hence, we argue that the optimal TSP solution depends solely on the spatial distribution of the nodes in the graph and is independent of any partially constructed tour. While previous works attempt to incorporate such context, we demonstrate in this work that the context from a partially constructed TSP tour is not necessary for building optimal solutions.

Most works define the TSP solution as a sequence of nodes, but Kwon et al. (2020) point out that this can lead to multiple representations of the same underlying solution. The optimal solution to a TSP-$n$ problem (instance with $n$ nodes) can have $2n$ representations. For example with $n = 5$, $\tau = (v_1, v_2, v_3, v_4, v_5)$ and $\tau' = (v_2, v_3, v_4, v_5, v_1)$ represent the same solution, and there are total 5 such representations starting with each node. Again $\tau'' = (v_5, v_4, v_3, v_2, v_1)$ in the reverse direction also represents the same solution and there are again 5 such reverse representations corresponding to each of the aforementioned representations.

To avoid this problem, we represent the TSP solution as an adjacency list. For the above example, $\tau$ can be represented as:

$$v_1 : \{v_2, v_5\}$$
$$v_2 : \{v_1, v_3\}$$
$$v_3 : \{v_2, v_4\}$$
$$v_4 : \{v_3, v_5\}$$
$$v_5 : \{v_1, v_4\}$$

As the same optimal tour can be produced in clockwise and anti-clockwise directions, the optimal entry and exit edges for every city are interchangeable. Therefore, instead of predicting the tour step by step, our equivalent objective is to predict these two optimal edges (entry and exit) for every node directly, without the context of any partially constructed tour. This greatly speeds up solution generation, as the predictions can be performed in parallel for each node.

We adopt Self-Improvement Learning (SIL), an unsupervised framework that iteratively refines predictions through self-generated pseudo-labels. Trained as a unified model across varying graph sizes, our approach achieves near-optimal performance on benchmark instances, especially on smaller graphs, while delivering competitive performance with other approaches on larger graphs.

Our contributions are threefold: (1) We demonstrate that it is possible to generate high-quality TSP solutions, in a highly efficient parallel manner, without requiring context of partially constructed tours. (2) We show that this approach is not only feasible but also learnable from scratch using unsupervised data, without costly expert supervision. (3) We introduce new metrics that emphasize the raw performance of the neural model without post-processing, addressing concerns by Xia et al. (2024), and show that our model outperforms previous works on these metrics.

## 2 RELATED WORK

### 2.1 AUTOREGRESSIVE METHODS

A prominent approach in neural TSP solving, the autoregressive (AR) framework constructs solutions sequentially by selecting one city at a time. Early pioneering work by Vinyals et al. (2015) introduced the Pointer Network, which employs recurrent neural networks (RNNs) to decode TSP solutions autoregressively under supervised learning. This framework was subsequently enhanced by Bello et al. (2017), through reinforcement learning (RL) to eliminate the dependency on pre-computed optimal solutions, using negative tour length as a reward signal in an actor-critic architecture.

The Transformer architecture (Vaswani et al., 2017) was introduced to this domain by Kool et al. (2019) through the Attention Model (AM) that leverages self-attention mechanisms for improved performance across multiple combinatorial problems including TSP and CVRP. Building upon AM, Kwon et al. (2020) introduced Policy Optimization with Multiple Optima (POMO), which exploits the symmetric nature of TSP by training on multiple starting points simultaneously.

Subsequent developments have refined these approaches through various enhancements: multiple decoders (Xin et al., 2021), matrix encoding (Kwon et al., 2021), heterogeneous attention mechanisms (Li et al., 2021), and symmetry-aware training (Kim et al., 2022). Advanced variants include heavy decoder architectures (Luo et al., 2023) that dynamically re-embed node representations at each decoding step, achieving impressive performance on large-scale instances at the cost of computational efficiency.

Despite their success, autoregressive methods face inherent limitations: they require multiple forward passes through the neural network, suffer from error propagation in long sequences, and exhibit high training variance on large graphs (Bengio et al., 2021).

### 2.2 NON-AUTOREGRESSIVE METHODS

Non-autoregressive (NAR) approaches address the computational bottlenecks of sequential construction by generating solution representations in a single forward pass. Joshi et al. (2019) introduced *heatmaps* — probability distributions over edges indicating their likelihood of appearing in the optimal tour. Their Graph Convolutional Network (GCN)-based approach demonstrated the potential of direct edge prediction for TSP solving.

Recent advances in NAR methods have explored various architectural and training innovations. Fu et al. (2021) enhanced the heatmap approach with graph partitioning, where sub-heatmaps are created and merged back to build the complete heatmap for the entire graph. Qiu et al. (2022) proposed DIMES, employing RL with efficient sampling strategies for REINFORCE-based gradient estimation. Sun & Yang (2023) introduced DIFUSCO, a diffusion-based model that treats TSP as iterative denoising of edge selection variables, requiring fully supervised training on solutions at each scale. However, the denoising scheme of diffusion-based methods also requires multiple passes through the neural network during inference, similar to AR approaches, limiting their effectiveness.

A critical limitation of existing NAR methods is their reliance on sophisticated post-processing techniques to achieve competitive performance. These approaches typically require Monte Carlo Tree Search (MCTS) (Fu et al., 2021; Qiu et al., 2022), beam search, active search (Hottung et al., 2022), or local optimization procedures such as 2-opt (Sun & Yang, 2023; Min et al., 2023) to convert heatmaps into valid TSP tours. While these post-processing steps improve solution quality, they significantly increase inference time and, as noted by Xia et al. (2024), potentially overshadow the neural model's intrinsic capabilities.

### 2.3 TRAINING PARADIGMS

Training strategies for neural TSP solvers span supervised, unsupervised, and reinforcement learning paradigms. Supervised approaches (Joshi et al., 2019; Sun & Yang, 2023) require expensive optimal or near-optimal solutions as labels, limiting scalability to larger instances. Reinforcement learning methods (Bello et al., 2017; Kool et al., 2019; Kwon et al., 2020; Qiu et al., 2022) optimize through environment interaction but often suffer from high variance and training instability. Min et al. (2023)

demonstrated unsupervised learning for heatmap generation using geometric scattering-based networks with a heuristically designed loss function.

Self-improvement learning has emerged as a promising alternative (Corsini et al., 2024; Pirnay & Grimm, 2024; Luo et al., 2025), iteratively improving model predictions through self-generated pseudo-labels, which avoids the complexity of RL and the need for designing a reward function.

## 3 METHODOLOGY

### 3.1 MODEL ARCHITECTURE

Our architecture follows the standard Transformer design from Vaswani et al. (2017), which was first used by Kool et al. (2019) for solving the TSP. The details are described below:

#### 3.1.1 INPUT REPRESENTATION AND EMBEDDING

Let a TSP instance be defined by a set of $N$ nodes, where each node $i$ possesses features $x_i \in \mathbb{R}^{d_{in}}$. For standard 2D TSP instances, $d_{in} = 2$. The input features $X \in \mathbb{R}^{N \times 2}$ are first processed by a multi-layer perceptron (MLP) to embed them into a higher-dimensional space of dimension $d_h$. This initial embedding network allows the model to learn richer, non-linear feature representations of individual nodes before contextualization. We use a 3 layer MLP which is defined as:

$$H^{(0)} = \text{MLP}(X) = \text{ReLU}(\text{ReLU}(XW_1 + b_1)W_2 + b_2)W_3 + b_3 \tag{1}$$

where $W_k$ and $b_k$ are learned parameters of the network layers. The resulting embedding matrix $H^{(0)} \in \mathbb{R}^{N \times d_h}$ serves as the input to the Transformer encoder.

#### 3.1.2 TRANSFORMER ENCODER

The encoder, composed of a stack of $L$ identical layers, refines the node embeddings by capturing the global context of the graph structure. Each encoder layer contains two primary sub-layers: multi-head self-attention (MHA) and a feed-forward network (FFN). Residual connections and layer normalization are applied around each sub-layer. For an input $H^{(l-1)}$ to layer $l$, the computation proceeds as:

$$A^{(l)} = \text{LayerNorm}(H^{(l-1)} + \text{MHA}(H^{(l-1)})) \tag{2}$$

$$H_{enc}^{(l)} = \text{LayerNorm}(A^{(l)} + \text{FFN}(A^{(l)})) \tag{3}$$

The MHA mechanism computes scaled dot-product attention, allowing each node to aggregate information from all other nodes in the graph:

$$\text{MHA}(H) = \text{Concat}(\text{head}_1, \ldots, \text{head}_h)W^O, \quad \text{where head}_i = \text{Attention}(HW_i^Q, HW_i^K, HW_i^V) \tag{4}$$

The final output of the encoder stack, $H_{enc} \in \mathbb{R}^{N \times d_h}$, represents a set of contextually-rich node embeddings.

#### 3.1.3 TRANSFORMER DECODER

The decoder has an additional cross-attention sub-layer, as in the standard Transformer. The decoder stack also consists of $L$ layers. We feed the final encoder output $H_{enc}$ as the initial input sequence to the decoder, i.e., $H_{dec}^{(0)} = H_{enc}$. Within each decoder layer $l$, the cross-attention mechanism uses queries generated from the output of the decoder's self-attention sub-layer, while keys and values are supplied by the encoder output $H_{enc}$:

$$A_{self}^{(l)} = \text{LayerNorm}(H_{dec}^{(l-1)} + \text{MHA}_{self}(H_{dec}^{(l-1)})) \tag{5}$$

$$A_{cross}^{(l)} = \text{LayerNorm}(A_{self}^{(l)} + \text{MHA}_{cross}(Q = A_{self}^{(l)}, K = H_{enc}, V = H_{enc})) \tag{6}$$

$$H_{dec}^{(l)} = \text{LayerNorm}(A_{cross}^{(l)} + \text{FFN}(A_{cross}^{(l)})) \tag{7}$$

### 3.1.4 OUTPUT PROBABILITY CALCULATION

We first compute the *compatibility*, which we interpret as unnormalized log-probabilities (logits), similar to Kool et al. (2019) for calculation of output probabilities. The compatibility is computed by combining the outputs from the encoder and decoder where the final decoder output $H_{dec} = H_{dec}^{(L)}$ serves as the query representations, and the final encoder output $H_{enc}$ serves as key representations:

$$U = \frac{H_{dec}(H_{enc})^T}{\sqrt{d_h}} \tag{8}$$

The diagonal elements of the resulting matrix are masked to prevent self-loops ($U_{ii} = -\infty$). A softmax function is then applied row-wise to obtain the probability matrix $P$, where $P_{ij}$ represents the likelihood of optimal edge from node $i$ to node $j$:

$$P = \text{softmax}(U) \tag{9}$$

## 3.2 TRAINING PIPELINE

The model is trained using a combination of data augmentation and self-improvement learning. The training pipeline is described next and visualized in Figure 1.

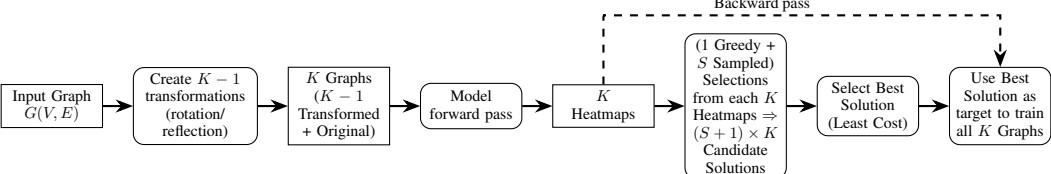

Figure 1: Self-improvement training pipeline with geometric augmentation

### 3.2.1 GEOMETRIC TRANSFORMATION OF GRAPHS

Previous works (Kwon et al., 2020; Kim et al., 2022) have observed that TSP solutions remain unchanged under rotation and reflection. To exploit this fact, we apply geometric transformations to each training instance and generate $K - 1$ augmented versions. First, we centre the graph by subtracting its centroid from the node coordinates. Next, we create $K - 1$ copies, apply reflection across the $y$-axis to half of them, followed by random rotations to all $K - 1$ copies, resulting in $K$ total versions of each graph. This leads to a couple of advantages: (1) It enables the model to learn representations that are invariant to rotation and reflection, which improves generalization as the model learns to produce consistent solution across different geometric orientations of the same underlying problem structure. (2) By sharing the pool of candidate solutions across the $K$ versions for finding the training target for SIL, computational efficiency is significantly improved.

### 3.2.2 SEQUENTIAL EDGE SELECTION

Traditional TSP solvers construct tours by sequentially selecting nodes. In contrast, we adopt an alternative approach which builds the tour by sequentially selecting edges, adding them to the solution set in any order provided no conflicts arise. The construction algorithm first calculates edge probabilities $Q_{ij} = P_{ij} \cdot P_{ji}$ for undirected edges. It then iteratively selects edges based on these probabilities, subject to two constraints: (1) the degree of any node must not exceed 2, and (2) no cycles are formed prematurely. Empirical evaluations (reported in Table 1) demonstrate that this method yields more accurate results compared to the conventional node-by-node construction, achieving better results with significantly fewer sampled trajectories. This construction method is detailed in Algorithm 1.

Notably, edge-by-edge construction is not possible with autoregressive (AR) models, as it requires upfront probabilities for all graph edges, whereas AR models compute probabilities only for edges incident to the current node at each step.

---

**Algorithm 1** Sequential Edge Selection for TSP tour construction

---

**Input:** Probability matrix $P \in [0,1]^{N \times N}$ (row-normalized), selection mode (greedy or sampling)
**Output:** Tour as a set of $N$ undirected edges
 1: Symmetrize probabilities: $Q_{ij} \leftarrow P_{ij} \cdot P_{ji}$
 2: Let $\mathcal{E}$ be the set of all possible $\binom{N}{2}$ undirected edges $\{i, j\}$ with $i < j$
 3: Initialize empty tour edge list $\tau = \emptyset$
 4: Initialize node degrees $d_v = 0$ for all $v \in \{0, \ldots, N-1\}$
 5: Initialize connected components as singletons $\{\{0\}, \{1\}, \ldots, \{N-1\}\}$
 6: Initialize processed indicators $I_e = $ False for all $e \in \mathcal{E}$
 7: **while** $|\tau| \leq N$ **do**
 8:     **if** selection mode is greedy **then**
 9:        Select edge $e$ with the highest $Q_e$ among edges where $I_e = $ *False*
10:     **else if** selection mode is sampling **then**
11:        Sample edge $e$ from the distribution defined by $Q$ over edges where $I_e = $ *False*
12:     **end if**
13:     Add $e = \{u, v\}$ to $T$
14:     Set $I_e \leftarrow$ *True*
15:     Increment degrees: $d_u \leftarrow d_u + 1$, $d_v \leftarrow d_v + 1$
16:     Set $I_{e'} \leftarrow$ *True* for all edges $e'$ incident to any node $w \in \{u, v\}$ where $d_w = 2$
17:     Merge the connected components containing $u$ and $v$ into a new connected component
18:     **if** number of edges with $I_e = $ *False* $> 1$ **then**
19:        Set $I_{e'} \leftarrow$ *True* for all edges $e'$ formed by nodes within this new connected component
20:     **end if**
21: **end while**
22: **return** the constructed tour edge list $\tau$

---

### 3.2.3 SELF-IMPROVEMENT LEARNING

**1. Candidate Tour Generation**  For each input graph and its $K - 1$ augmentations, the model outputs the probability matrix (heatmap) $P$. We then generate a set of candidate solutions for each $P$, using the edge-by-edge tour construction method described in Algorithm 1.

We can generate solutions using different modes: (1) **Greedy:** Sequentially select the highest probability valid edge. (2) **Sampling:** Sequentially sample from valid edges according to their probabilities given by $Q$. We use a combined approach by generating 1 greedy solution and $S = 16$ sampled solutions per graph.

**2. Pseudo-Label Generation**  After generating a pool of candidate tours, we evaluate each tour by calculating the tour length. The edges of the tour $\tau^*$ with the minimum length are selected as the pseudo-labels for training.

$$\tau^* = \underset{\tau \in \{\tau_{greedy}, \tau_{sample}^{(1)}, \ldots, \tau_{sample}^{(n)}\}}{\arg\min} \text{Cost}(\tau) \tag{10}$$

**3. Loss function**  The model parameters $\theta$ are updated by minimizing the negative log-likelihood of generating the best-found tour $\tau^*$. This is similar to supervised learning, but the model's output is used to generate the pseudo-labels. As the training progresses and the model improves, so does the quality of pseudo-labels, creating a virtuous cycle and enabling it to discover even better solutions in subsequent iterations.

$$\mathcal{L}(\theta) = -\mathbb{E}_{G \sim \mathcal{D}} \left[ \frac{1}{2 \cdot |\tau|} \sum_{(i,j) \in \tau^*} \log P_{ij}(G; \theta) + \log P_{ji}(G; \theta) \right] \tag{11}$$

### 3.3 INFERENCE STRATEGY

For the TSP, we know that each node must be connected with exactly two distinct edges in the solution trajectory. But for each node, we cannot select both the edges simultaneously, as it can lead

to the same edge being selected twice. Therefore, the two edges must be selected one after the other, *in two selections*.

Since the probability mass of each row in the heatmap sums up to 1, after the first optimal edge has been selected with probability mass $p$, the next edge has to be selected after excluding the selected edge and normalizing the remaining probability mass $(1 - p)$ between the rest of the edges. If the two optimal edges are assigned probability $p$ and $1 - p$, where $p > 0$, then the probability that each of these edges is selected, after two selections, is 100%, independent of $p$. This ensures that if the probability mass is concentrated on two edges, they are selected with certainty.

We interpret the model output $P_{i,j}$ to mean the probability that the edge $e_{i,j}$ is selected in the first selection. Then the probability that $e_{i,j}$ will be selected in the second selection:

$$\sum_{k \neq j} P_{i,k} \cdot \frac{P_{i,j}}{1 - P_{i,k}} = P_{i,j} \sum_{k \neq j} \frac{P_{i,k}}{1 - P_{i,k}} = P_{i,j} \left( -\frac{P_{i,j}}{1 - P_{i,j}} + \sum_k \frac{P_{i,k}}{1 - P_{i,k}} \right) \quad (12)$$

Note that the term $S_i = \sum_k \frac{P_{i,k}}{1 - P_{i,k}}$, is independent of $j$, which means this needs to be computed only once per row, and reused for efficiency.

The probability $\hat{P}_{i,j}$ that edge $e_{i,j}$ will be selected (after two selections):

$$\hat{P}_{i,j} = p(e_{i,j} \text{ picked in first selection}) + p(e_{i,j} \text{ picked in second selection}) \quad (13)$$

$$= P_{i,j} + P_{i,j} \left( -\frac{P_{i,j}}{1 - P_{i,j}} + S_i \right) \quad (14)$$

$$= P_{i,j} \left( 1 - \frac{P_{i,j}}{1 - P_{i,j}} + S_i \right) \quad (15)$$

The joint probability $\hat{Q}_{i,j}$ for the undirected edge between $i$ and $j$ i.e. the probability that edges $e_{i,j}$ *and* $e_{j,i}$ both will be selected, after two selections:

$$\hat{Q}_{i,j} = \hat{P}_{i,j} \cdot \hat{P}_{j,i} \quad (16)$$

We now select the top two elements per row of the symmetrical heatmap $\hat{Q}$ and interpret it as the model's prediction of the TSP tour without further post-processing.

## 4 EXPERIMENTS

**Hyperparameters.** We use a balanced encoder-decoder architecture where both the encoder and decoder have $L = 12$ layers each. The attention layers have 8 heads, each with embedding dimension of 16 ($d_h = 8 \times 16 = 128$). The feed-forward layers have hidden dimension of $4 \times d_h = 512$. For each problem instance, 17 (1 greedy + 16 sampled) candidate solutions are generated following Algorithm 1.

**Training.** Unlike many prior works that train separate models specialized for each graph size, we train a single model across varying graph sizes to achieve near-optimal performance. We train the model from scratch using a progressive curriculum on increasing graph sizes in multiple phases:

- We initialize training on small graphs with $N = 20$ for a brief period (approximately 30 minutes). During this period, we ramp up the learning rate $\eta$ linearly from 0 to $5 \times 10^{-4}$.
- We continue training on medium-sized graphs with $N = 50$ for an extended duration (about 10 hours), until performance approaches optimality. We employ a Cosine Annealing with Warm Restarts schedule (Loshchilov & Hutter, 2017), with a maximum learning rate $\eta_{\max} = 5 \times 10^{-4}$.
- We then train on mixed graph sizes with $N \in \{50, 100\}$, randomly selecting $N$ per epoch. Since the graph size remains fixed during each epoch, we limit the number of batches trained per epoch to 16 batches, because the model would otherwise start overfitting to a specific graph size until the graph size changes in a subsequent epoch. At this stage, the maximum learning rate is set to $\eta_{\max} = 2 \times 10^{-4}$, using the same schedule.

- Subsequently, we progressively incorporate larger graph sizes into the mix, up to $N \in \{20, 50, 100, 200, 500\}$. From this point, we use $\eta_{\max} = 1 \times 10^{-4}$.

As the graph size changes every epoch, we dynamically adjust the batch size to maintain a constant number of tokens (nodes) per batch. Specifically, we set the number of nodes per batch (per GPU) to 6,400, resulting in a batch size (per GPU) of 128 for TSP50 and 64 for TSP100. This approach ensures that GPU memory requirements remain roughly consistent across epochs, irrespective of graph size. Training is conducted on $4\times$ RTX 2080 Ti GPUs in parallel, yielding an effective batch size of $4\times$ the aforementioned per-GPU values.

We set $K = 2$, which means only one additional geometrically transformed version is used. We avoid using larger values of $K$ because during experimentation we found that the model learns faster from more diverse data in the batch.

We use Adam (Kingma & Ba, 2014) as the optimizer for training through back-propagation.

**Baselines.** We compare our approach with: (1) Classical Solver: Concorde (Applegate et al., 2006) (2) Autoregressive (AR) methods: AM (Kool et al., 2019), POMO (Kwon et al., 2020), BQ-NCO (Drakulic et al., 2023), LEHD (Luo et al., 2023) (3) Non-autoregressive (NAR) methods: Att-GCN (Fu et al., 2021), DIMES Qiu et al. (2022), DIFUSCO (Sun & Yang, 2023), UTSP (Min et al., 2023), SoftDist (Xia et al., 2024)

**Testing.** We evaluate on Euclidean TSP instances (uniform $[0,1]^2$ coordinates), following previous works. Test datasets for TSP $N \in \{20, 50, 100\}$ (10k test instances each) are from Kool et al. (2019), and for TSP $N \in \{200, 500\}$ (128 test instances each) are taken from Fu et al. (2021).

## 4.1 RESULTS

Table 1: Results on TSP using conventional sequential tour construction. Gaps are with respect to Concorde. G and S indicate Greedy and Sampled decoding using sequential node by node construction, respectively. GE and SE indicate Greedy and Sampled decoding using Algorithm 1, respectively. Baseline results for AR methods are sourced from original papers, while the results for NAR methods are obtained by running our own analysis on heatmaps made publicly available by the authors of the respective papers.

| METHOD | TYPE | TSP-20 Len. ↓ | Gap ↓ | Time ↓ | TSP-50 Len. ↓ | Gap ↓ | Time ↓ | TSP-100 Len. ↓ | Gap ↓ | Time ↓ | TSP-200 Len. ↓ | Gap ↓ | Time ↓ | TSP-500 Len. ↓ | Gap ↓ | Time ↓ |
|---|---|---|---|---|---|---|---|---|---|---|---|---|---|---|---|---|
| Concorde | Exact Solver | 3.831 | 0.00% | 2.31m | 5.692 | 0.00% | 13.68m | 7.764 | 0.00% | 1.04h | 10.728 | 0.00% | 3.44m | 16.584 | 0.00% | 37.66m |
| AM | RL+G | 3.841 | 0.287% | 6.03s | 5.785 | 1.657% | 34.92s | 8.101 | 4.38% | 1.83m | 11.610 | 8.308% | 5s | 20.019 | 20.990% | 1.51m |
| POMO | RL+G | 3.83 | 0.12% | ≪ 1s | 5.73 | 0.64% | 1s | 7.84 | 1.07% | 2s | — | — | — | — | — | — |
| LEHD | SL+G | — | — | — | — | — | — | — | 0.577% | 0.4m | — | 0.859% | 3s | — | 1.560% | 0.3m |
| BQ-NCO | SL+G | — | — | — | — | — | — | — | 0.35% | 2m | — | 0.54% | 9s | — | **1.18%** | 55s |
| AM | RL+S | 3.832 | 0.05% | 16.47m | 5.719 | 0.491% | 22.85m | 7.974 | 2.739% | 1.23h | 11.45 | 6.816% | 4.49m | 22.641 | 36.838% | 15.64m |
| POMO | RL+S | 3.83 | 0.04% | ≪ 1s | 5.70 | 0.21% | 1s | 7.80 | 0.46% | 11s | — | — | — | — | — | — |
| UTSP | UL+G | 4.569 | 19.339% | 0.01s | 7.323 | 28.717% | 0.02s | 10.558 | 36.014% | 0.06 | 14.752 | 37.547% | 0.03s | 25.547 | 54.055% | 0.17s |
| Att-GCN | SL+G | 3.884 | 1.379% | 0.01s | 7.277 | 27.862% | 0.02s | 10.430 | 34.330% | 0.06 | 16.409 | 52.936% | 0.03s | 31.396 | 89.325% | 0.17s |
| DIMES | RL+G | — | — | — | — | — | — | — | — | — | — | — | — | 51.442 | 210.219% | 0.17s |
| DIFUSCO ×1 | SL+G | — | — | — | 6.089 | 6.924% | 0.02s | 9.348 | 20.349% | 0.06s | — | — | — | 24.459 | 47.484% | 0.17s |
| DIFUSCO ×50 | SL+G | — | — | — | 5.768 | 1.334% | 0.02s | 8.414 | 8.350% | 0.07s | — | — | — | 18.800 | 13.366% | 0.18s |
| SoftDist | SoftDist+G | 4.788 | 24.940% | 0.01s | 7.022 | 23.367% | 0.02s | 12.772 | 64.491% | 0.06s | 19.560 | 82.350% | 0.03s | 43.818 | 164.237% | 0.17s |
| **Ours** | SIL+G | **3.835** | **0.125%** | 0.01s | **5.743** | **0.892%** | 0.02s | **7.975** | **2.710%** | 0.06s | **11.315** | **5.458%** | 0.03s | **18.790** | **13.306%** | 0.18s |
| UTSP | UL+S | 4.560 | 19.093% | 0.01s | 7.323 | 28.716% | 0.06s | 10.558 | 36.014% | 0.20s | 14.752 | 37.547% | 0.07s | 25.547 | 54.055% | 0.28s |
| Att-GCN | SL+S | 3.839 | 0.224% | 0.01s | 7.237 | 27.172% | 0.06s | 10.421 | 34.211% | 0.20s | 16.409 | 52.936% | 0.07s | 31.396 | 89.325% | 0.27s |
| DIMES | RL+S | — | — | — | — | — | — | — | — | — | — | — | — | 50.751 | 206.047% | 0.27s |
| DIFUSCO ×1 | SL+S | — | — | — | 5.798 | 1.843% | 0.06s | 8.902 | 14.609% | 0.20s | — | — | — | 24.459 | 47.484% | 0.28s |
| DIFUSCO ×50 | SL+S | — | — | — | 5.701 | 0.164% | 0.06s | 7.980 | 2.769% | 0.20s | — | — | — | 18.800 | 13.366% | 0.28s |
| SoftDist | SoftDist+S | 4.184 | 9.180% | 0.01s | 7.022 | 23.367% | 0.06s | 11.436 | 47.295% | 0.20s | 17.979 | 67.600% | 0.08s | 40.799 | 146.041% | 0.27s |
| **Ours** | SIL+S | **3.831** | **0.001%** | 0.01s | **5.697** | **0.095%** | 0.06s | **7.826** | **0.783%** | 0.20s | **11.067** | **3.153%** | 0.07s | **18.737** | **12.983%** | 0.28s |
| UTSP | UL+GE | 4.356 | 13.738% | 0.03s | 6.760 | 18.794% | 0.14s | 9.367 | 20.652% | 1.06s | 13.060 | 21.744% | 0.18s | 21.800 | 31.461% | 1.73s |
| Att-GCN | SL+GE | 3.845 | 0.377% | 0.03s | 6.490 | 14.029% | 0.14s | 9.089 | 17.051% | 1.07s | 13.641 | 27.124% | 0.18s | 24.827 | 49.712% | 1.73s |
| DIMES | RL+GE | — | — | — | — | — | — | — | — | — | — | — | — | 66.981 | 303.917% | 1.74s |
| DIFUSCO ×1 | SL+GE | — | — | — | 5.789 | 1.683% | 0.14s | 8.314 | 7.048% | 1.06s | — | — | — | 20.139 | 21.429% | 1.74s |
| DIFUSCO ×50 | SL+GE | — | — | — | 5.708 | 0.284% | 0.14s | 8.035 | 3.479% | 1.07s | — | — | — | 18.283 | 10.244% | 1.74s |
| SoftDist | SoftDist+GE | 4.502 | 17.442% | 0.03s | 6.485 | 13.917% | 0.14s | 11.643 | 49.940% | 1.06s | 17.374 | 61.945% | 0.19s | 38.569 | 132.610% | 1.73s |
| **Ours** | SIL+GE | **3.831** | **0.011%** | 0.03s | **5.698** | **0.010%** | 0.14s | **7.797** | **0.415%** | 1.07s | **10.877** | **1.385%** | 0.18s | **17.585** | **6.034%** | 1.73s |
| UTSP | UL+SE | 4.353 | 13.670% | 0.17s | 6.760 | 18.794% | 2.37s | 9.367 | 20.652% | 19.42s | 13.060 | 21.744% | 2.02s | 21.800 | 31.461% | 30.15s |
| Att-GCN | SL+SE | 3.833 | 0.069% | 0.17s | 6.434 | 13.049% | 2.38s | 9.065 | 16.740% | 19.43s | 13.633 | 27.052% | 2.03s | 24.827 | 49.712% | 30.17s |
| DIMES | RL+SE | — | — | — | — | — | — | — | — | — | — | — | — | 64.370 | 288.148% | 30.15s |
| DIFUSCO ×1 | SL+SE | — | — | — | 5.707 | 0.259% | 2.38s | 7.974 | 2.687% | 19.43s | — | — | — | 19.882 | 19.887% | 30.14s |
| DIFUSCO ×50 | SL+SE | — | — | — | 5.700 | 0.146% | 2.39s | 7.905 | 1.800% | 19.43s | — | — | — | 17.777 | 7.195% | 30.15s |
| SoftDist | SoftDist+SE | 4.052 | 5.732% | 0.17s | 6.475 | 13.738% | 2.38s | 10.362 | 33.450% | 19.44s | 15.797 | 47.252% | 2.03s | 36.020 | 117.219% | 30.14s |
| **Ours** | SIL+SE | **3.830** | **-0.003%** | 0.17s | **5.692** | **0.010%** | 2.37s | **7.772** | **0.095%** | 19.42s | **10.765** | **0.344%** | 2.02s | **17.048** | **2.797%** | 30.15s |

The results presented in Table 1 demonstrate that our approach achieves superior performance on standard benchmarks against both NAR and AR baselines even when following conventional sequential decoding schemes. Especially on smaller graphs, our model achieves near-optimal results.

## 4.2 Proposed Additional Metrics

To demonstrate the superior raw performance of our model, we introduce additional metrics, which do not depend on advanced post-processing. These metrics measure model performance when edge predictions are considered following the scheme outlined in Section 3.3:

- **Optimal Edges**: Percentage of predicted edges in test dataset that match Concorde's prediction on the same dataset.
- **Valid Tours**: Percentage of graphs in test dataset where the predicted edges form a valid TSP solution.
- **Optimal Tours**: Percentage of graphs in test dataset where all the predicted edges are optimal i.e. the predictions match Concorde's solution for the graph.

Table 2: Comparison of heatmap based methods on metrics introduced in Section 4.2

| Method | TSP-20 | | | TSP-50 | | | TSP-100 | | | TSP-200 | | | TSP-500 | | |
| | Optimal Edges ↑ | Valid Tours ↑ | Optimal Tours ↑ | Optimal Edges ↑ | Valid Tours ↑ | Optimal Tours ↑ | Optimal Edges ↑ | Valid Tours ↑ | Optimal Tours ↑ | Optimal Edges ↑ | Valid Tours ↑ | Optimal Tours ↑ | Optimal Edges ↑ | Valid Tours ↑ | Optimal Tours ↑ |
|---|---|---|---|---|---|---|---|---|---|---|---|---|---|---|---|
| UTSP | 62.060% | 0% | 0% | 62.756% | 0% | 0% | 63.652% | 0% | 0% | 61.014% | 0% | 0% | 65.470% | 0% | 0% |
| Att-GCN | 96.451% | 72.490% | 65.790% | 80.316% | 0% | 0% | 82.060% | 0% | 0% | 79.480% | 0% | 0% | 77.151% | 0% | 0% |
| DIMES | — | — | — | — | — | — | — | — | — | — | — | — | 63.295% | 0% | 0% |
| DIFUSCO ×1 | — | — | — | 94.978% | 42.750% | 29.660% | 91.406% | 3.460% | 1.270% | — | — | — | **86.739%** | 0% | 0% |
| DIFUSCO ×50 | — | — | — | 94.283% | 88.600% | 40.340% | 89.369% | 41.880% | 3.610% | — | — | — | 83.595% | **2.344%** | 0% |
| SoftDist | 69.838% | 0% | 0% | 70.861% | 0% | 0% | 71.206% | 0% | 0% | 72.445% | 0% | 0% | 71.780% | 0% | 0% |
| **Ours** | **98.100%** | **98.230%** | **86.280%** | **96.199%** | **90.100%** | **53.730%** | **93.328%** | **70.240%** | **15.420%** | **89.470%** | **42.969%** | **0.781%** | 80.038% | 0.781% | 0% |

The proposed additional metrics highlight the strong raw performance capability of our approach. Our method achieves outstanding performance on the *Optimal Edges* metric, outperforming all other available baselines. We consistently outperform other baselines across graph sizes. While DIFUSCO manages to perform slightly better than us on the large TSP500 benchmark, it requires multiple passes through the model to generate the heatmaps, making it difficult to compare in a fair manner.

The *Valid Tours* metric showcases the ability to generate feasible solutions without post-processing. We achieve valid tours for a substantial share of graphs in the test dataset. However, it is not yet possible to guarantee valid TSP tours for *all* instances, as the model can fall short by a few edges even after correctly predicting most of the optimal edges. Therefore, as a fallback, we can still use any of the widely used construction methods (the results of which are given in Table 1) to construct a valid TSP tour based on the heatmap.

Our method achieves 98.230% valid tours on TSP20, significantly outperforming Att-GCN with only 72.490% valid tours. Although the performance of our model degrades as the the problem size increases, with 90.100% valid tours on TSP50 and 70.240% on TSP100, it still dramatically outperforms all other baselines where most other methods consistently fail to generate valid tours.

The *Optimal Tours* metric provides the most stringent evaluation, measuring complete solution optimality. Our approach achieves remarkable 86.280% optimal tours on TSP20, significantly better than Att-GCN's 65.790% and completely outperforming other methods that achieve 0% optimal tours. We maintain 53.730% optimal tours on TSP50 while all other methods achieve 0%, and we continue to achieve 15.420% optimal tours on TSP100 where baselines again fail completely. DIFUSCO once again manages slightly better result than us on TSP500, but other methods consistently fail to produce valid and optimal tours, while our approach demonstrates strong performance on smaller sizes and promising results on larger graphs.

## 5 Conclusion

In this work, we introduced a one-shot approach to solving the TSP through highly accurate heatmaps, eliminating the need for sequential tour construction and context from partial solutions. Trained with self-improvement learning without supervision across varying graph sizes, the model demonstrates the feasibility of solving the TSP efficiently without post-processing, addressing previously raised concerns about the validity of the NAR approach. Future research directions could include extending this approach to other routing problems, such as the Vehicle Routing Problem (VRP).

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

## A    CHOICE OF K FOR DATA AUGMENTATION

We found that the optimal value of $K$ — the total number of geometrically augmented versions per problem instance (including the original) — is $K = 2$. This was determined by examining the validation performance across different values of $K$, as shown in Figure 2. The plot clearly indicates that training without augmentation ($K = 1$) yields the worst performance. In contrast, the best results are achieved with the smallest degree of augmentation ($K = 2$). Adding more augmented copies beyond this point degrades performance, most likely because excessive repetition of the same underlying data reduces the effective information density available to the model during training.

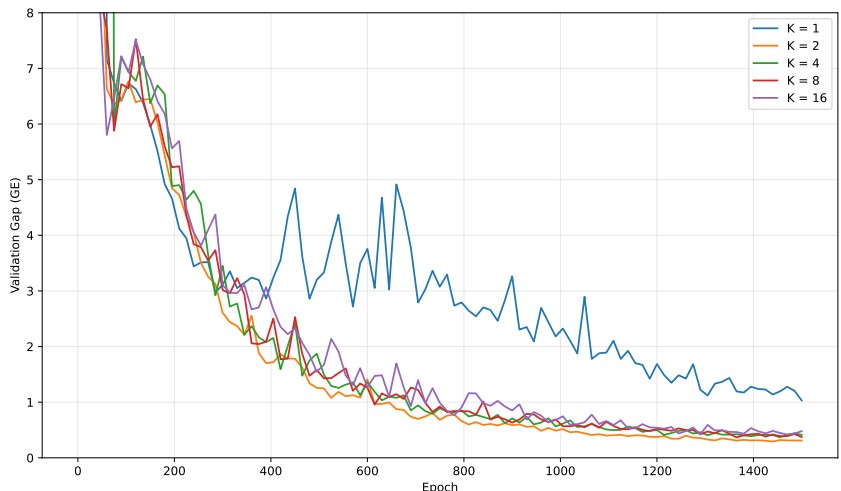

Figure 2: Validation Gap (GE) vs Training Epochs on TSP20 for different K values

## B    COMPARISON OF SELF-IMPROVEMENT LEARNING (SIL) WITH REINFORCEMENT LEARNING (RL) AND SUPERVISED LEARNING (SL)

**Supervised Learning:** The setup remains identical to our SIL setup, the only difference here is that the labels are not self-generated. Instead, we created a training dataset of 100,000 TSP20 instances, with optimal solutions generated by Concorde as labels.

**Reinforcement Learning:** We tried to train the model using the REINFORCE loss function as described by Kwon et al. (2020):

$$\nabla_\theta J(\theta) \approx \frac{1}{N} \sum_{i=1}^{N} (R(\boldsymbol{\tau}^i) - b^i(s)) \nabla_\theta \log p_\theta(\boldsymbol{\tau}^i|s) \tag{17}$$

where

$$p_\theta(\boldsymbol{\tau}^i|s) \equiv \prod_{t=2}^{M} p_\theta(a_t^i|s, a_{1:t-1}^i) \tag{18}$$

and

$$b^i(s) = b_{\text{shared}}(s) = \frac{1}{N} \sum_{j=1}^{N} R(\boldsymbol{\tau}^j) \quad \text{for all } i. \tag{19}$$

In our case, $p_\theta(a_t^i|s, a_{1:t-1}^i) = p_\theta(a_t^i|s)$, since for NAR methods such as ours, probability for all edges are output by the model in one-shot, without depending on partial tours. Therefore,

$$p_\theta(\boldsymbol{\tau}^i|s) \equiv \prod_{t=2}^{M} p_\theta(a_t^i|s) \tag{20}$$

However, we could not train our model with this due to instability. The gradient norm and validation cost would explode randomly after some period of training (see Figure 3). In contrast, SIL was extremely stable throughout training, with smooth convergence and no such failures. Therefore, SIL was the clear choice to train our model in an unsupervised manner.

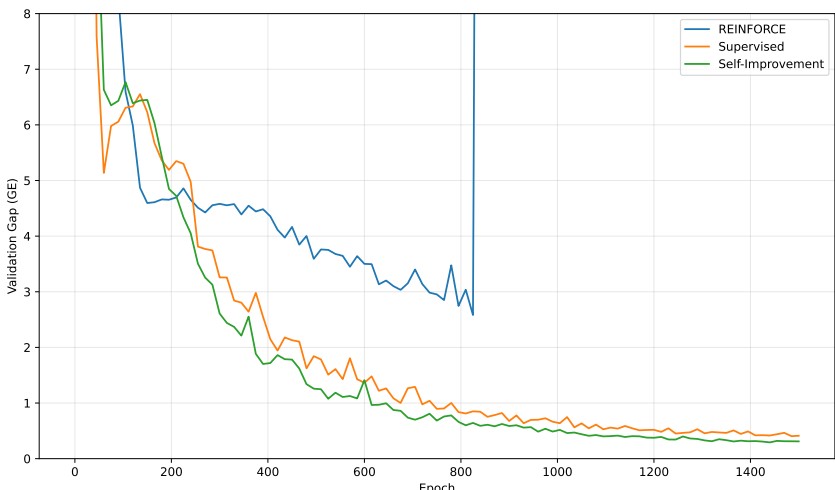

Figure 3: Validation Gap (GE) vs Training Epochs on TSP20 for different learning algorithms

It may seem counter-intuitive that Self-Improvement Learning (SIL) outperforms Supervised Learning (SL), given that SL is trained on ground-truth optimal labels, whereas SIL relies on noisier pseudo-labels which are suboptimal. The key reason is data scale: SL is constrained to a fixed dataset (100,000 labelled instances in this case), while SIL can generate and learn from a practically unlimited number of unlabelled instances with self-generated targets. This massive increase in effective training data more than compensates for the reduced label quality, leading to superior final performance.

## C  OUT-OF-DISTRIBUTION EVALUATIONS

### C.1  ON UNSEEN SIZES

For evaluation on problem sizes not seen during training, we evaluated our model on $N \in \{15, 30, 75, 150, 300, 600\}$. The results in Table 3 show excellent generalisation of our model across unseen sizes. The results are visualized in Figure 4.

### C.2  ON UNSEEN DISTRIBUTIONS

To evaluate the generalisation capability of our approach, we report the performance of our single unified model (which was trained only on uniform distribution) on three additional distributions from Fang et al. (2024): Clustered, Explosion and Implosion.

Additionally, we trained another model from scratch (with identical architecture and configuration as the initial model) on randomly generated TSP100 problems of all the four distributions (Uniform, Clustered, Explosion and Implosion). The results are reported in Table 4 and visualized in Figure 5.

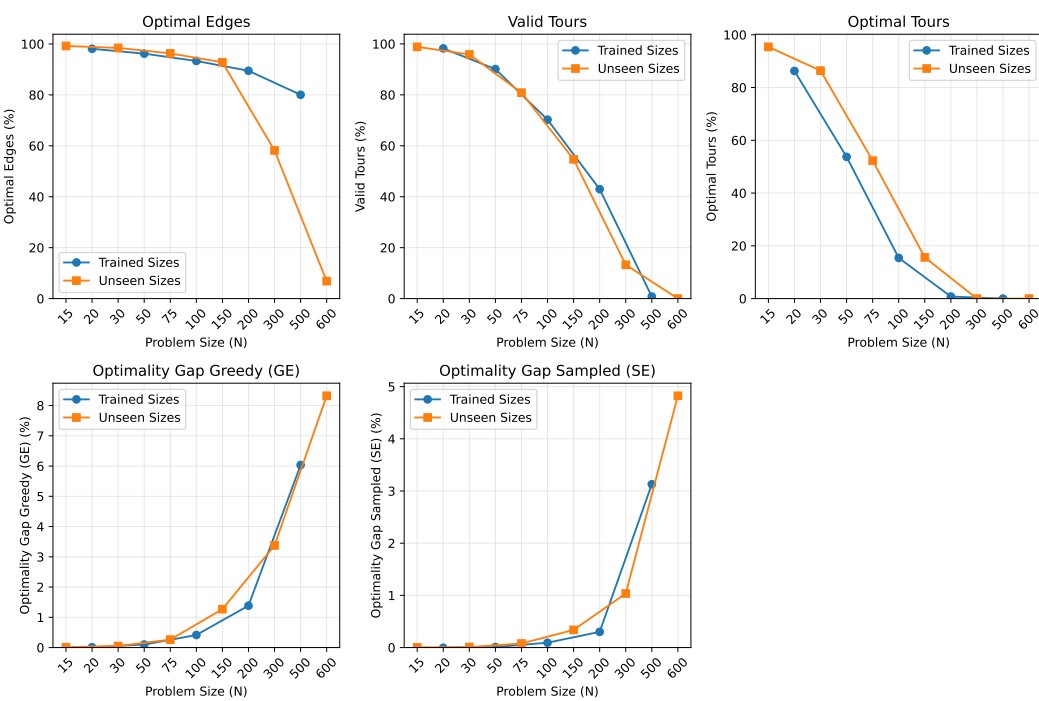

Figure 4: Evaluation on unseen problem sizes. Evaluation on trained sizes is given in blue for reference.

Table 3: Evaluation on unseen problem sizes of our single unified model trained on TSP-$N \in \{20, 50, 100, 200, 500\}$

| Metric | TSP15 | | TSP30 | | TSP75 | | TSP150 | | TSP300 | | TSP600 | |
|---|---|---|---|---|---|---|---|---|---|---|---|---|
| | Cost | Gap | Cost | Gap | Cost | Gap | Cost | Gap | Cost | Gap | Cost | Gap |
| G | 3.392 | 0.073% | 4.575 | 0.389% | 6.935 | 1.777% | 9.940 | 6.228% | 14.068 | 8.713% | 21.422 | 18.667% |
| S | 3.390 | 0.003% | 4.558 | 0.025% | 6.841 | 0.409% | 9.626 | 2.902% | 13.896 | 7.382% | 21.396 | 18.518% |
| GE | 3.390 | 0.013% | 4.559 | 0.051% | 6.831 | 0.262% | 9.473 | 1.269% | 13.378 | 3.376% | 19.554 | 8.323% |
| SE | 3.390 | 0.004% | 4.557 | 0.009% | 6.818 | 0.078% | 9.385 | 0.339% | 13.074 | 1.035% | 18.923 | 4.824% |
| G + 2-Opt | 3.390 | 0.004% | 4.558 | 0.022% | 6.820 | 0.096% | 9.383 | 0.316% | 13.042 | 0.787% | 18.468 | 2.304% |
| S + 2-Opt | 3.390 | 0.002% | 4.557 | 0.010% | 6.818 | 0.078% | 9.385 | 0.333% | 13.042 | 0.785% | 18.467 | 2.297% |
| GE + 2-Opt | 3.390 | 0.003% | 4.558 | 0.012% | 6.817 | 0.064% | 9.376 | 0.233% | 13.023 | 0.640% | 18.381 | 1.823% |
| SE + 2-Opt | 3.390 | 0.003% | 4.557 | 0.007% | 6.816 | 0.050% | 9.373 | 0.202% | 13.013 | 0.560% | 18.373 | 1.776% |
| Optimal Edges | 99.202% | | 98.450% | | 96.292% | | 92.794% | | 58.184% | | 6.840% | |
| Valid Tours | 98.850% | | 95.850% | | 80.850% | | 54.688% | | 13.281% | | 0% | |
| Optimal Tours | 95.430% | | 86.400% | | 52.300% | | 15.625% | | 0% | | 0% | |

The results show reasonable generalization from uniform training and even stronger performance when all distributions are included in training, confirming that our approach is not inherently limited to uniform Euclidean graphs.

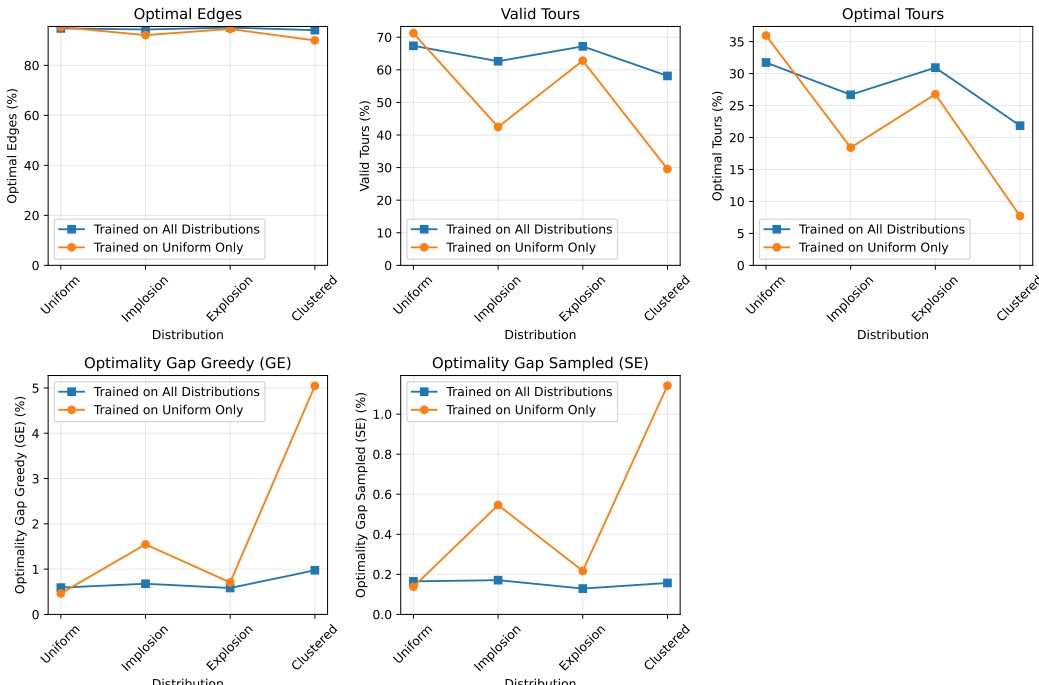

Figure 5: Out-of-distribution evaluation of model trained only on uniform distribution. Additional model trained on all distributions (shown in blue) for reference.

Table 4: Out-of-Distribution evaluation of our unified model (trained only on uniform distribution) on TSP100

| Distribution | Uniform | | Implosion | | Explosion | | Clustered | |
| --- | --- | --- | --- | --- | --- | --- | --- | --- |
| | Length | Gap | Length | Gap | Length | Gap | Length | Gap |
| G | 7.978 | 2.757% | 7.641 | 7.429% | 6.764 | 3.891% | 5.965 | 13.887% |
| S | 7.829 | 0.847% | 7.314 | 2.747% | 6.595 | 1.212% | 5.664 | 7.997% |
| GE | 7.799 | 0.463% | 7.232 | 1.546% | 6.562 | 0.706% | 5.519 | 5.045% |
| SE | 7.774 | 0.138% | 7.163 | 0.546% | 6.532 | 0.217% | 5.326 | 1.142% |
| G + 2-Opt | 7.775 | 0.151% | 7.151 | 0.347% | 6.532 | 0.215% | 5.300 | 0.589% |
| S + 2-Opt | 7.774 | 0.137% | 7.149 | 0.323% | 6.530 | 0.187% | 5.303 | 0.657% |
| GE + 2-Opt | 7.771 | 0.099% | 7.144 | 0.243% | 6.528 | 0.153% | 5.291 | 0.419% |
| SE + 2-Opt | 7.770 | 0.084% | 7.141 | 0.205% | 6.527 | 0.134% | 5.289 | 0.376% |
| Optimal Edges | 95.293% | | 92.096% | | 94.561% | | 89.971% | |
| Valid Tours | 71.210% | | 42.450% | | 62.800% | | 29.550% | |
| Optimal Tours | 35.960% | | 18.420% | | 26.770% | | 7.710% | |

## D    ILLUSTRATIVE EXAMPLES OF SUCCESS AND FAILURE CASES

### D.1    SUCCESS CASE

Figure 6 shows the best case scenario where the model directly predicts every edge optimally, forming an optimal tour. This validates the idea that TSP solutions can be generated in one-step, without any sequential tour construction or post-processing.

Table 5: Evaluation of our model trained from scratch on TSP100 mixed distribution (all four distributions)

| Distribution | Uniform | | Implosion | | Explosion | | Clustered | |
| --- | --- | --- | --- | --- | --- | --- | --- | --- |
| | Length | Gap | Length | Gap | Length | Gap | Length | Gap |
| G | 8.009 | 3.155% | 7.399 | 3.818% | 6.726 | 3.192% | 5.538 | 5.138% |
| S | 7.840 | 0.987% | 7.211 | 1.174% | 6.576 | 0.870% | 5.341 | 1.353% |
| GE | 7.809 | 0.592% | 7.175 | 0.677% | 6.556 | 0.583% | 5.321 | 0.974% |
| SE | 7.776 | 0.165% | 7.139 | 0.171% | 6.527 | 0.129% | 5.279 | 0.157% |
| G + 2-Opt | 6.778 | 0.185% | 7.142 | 0.207% | 6.528 | 0.155% | 5.280 | 0.181% |
| S + 2-Opt | 7.776 | 0.163% | 7.139 | 0.172% | 6.527 | 0.133% | 5.280 | 0.172% |
| GE + 2-Opt | 7.773 | 0.127% | 7.136 | 0.130% | 6.525 | 0.105% | 5.277 | 0.123% |
| SE + 2-Opt | 7.771 | 0.105% | 7.135 | 0.107% | 6.524 | 0.083% | 5.276 | 0.104% |
| Optimal Edges | 94.781% | | 94.355% | | 95.106% | | 94.088% | |
| Valid Tours | 67.360% | | 62.620% | | 67.190% | | 58.120% | |
| Optimal Tours | 31.730% | | 26.690% | | 30.930% | | 21.850% | |

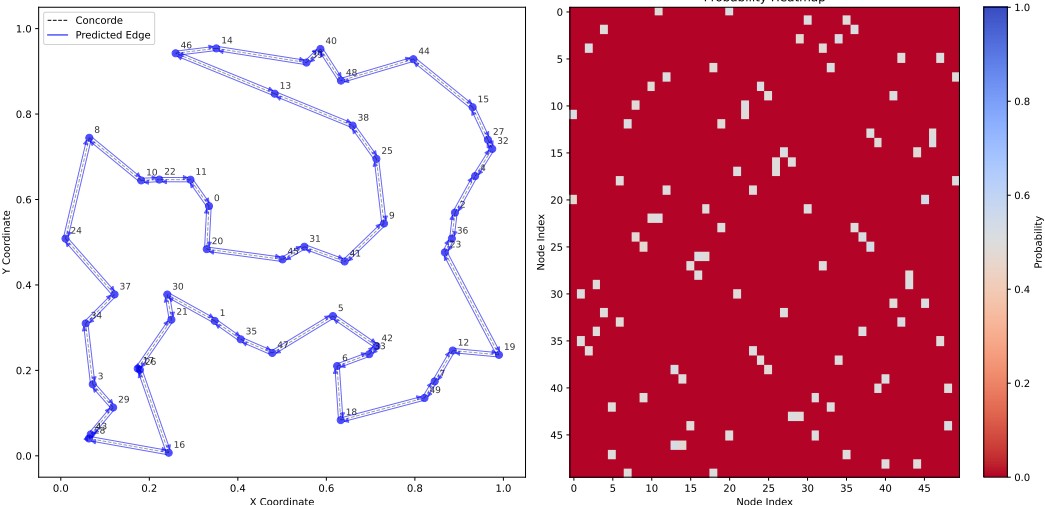

Figure 6: Example of success case in TSP50

## D.2 FAILURE CASE

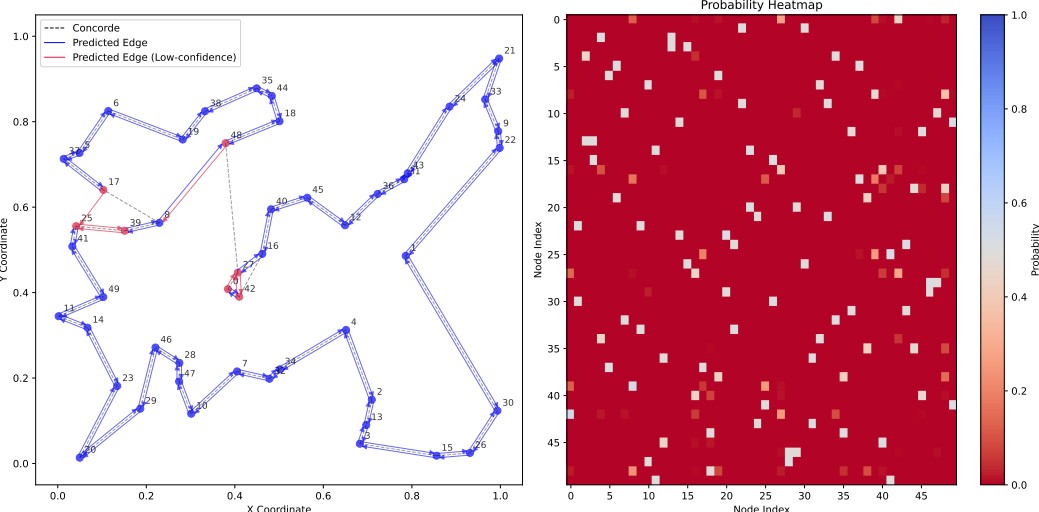

Figure 7: Example of failure case in TSP50

Figure 7 shows a failure case where the model fails to directly predict edges which form a valid Hamiltonian cycle. Valid TSP tours are not guaranteed by directly taking the top-2 predictions per node, but we can still use one of the sequential tour construction methods as fallback to generate valid solutions in such cases. This approach remains highly efficient as it avoids additional forward passes through the model and instead leverages the existing output to produce valid TSP solutions even in failure scenarios.

Figure 8 shows the Greedy Sequential Edge construction using the same heatmap from the model to construct a valid TSP, and Figure 9 shows the best trajectory found after generating 17 (1 Greedy + 16 sampled) tours using the Sequential Edge construction.

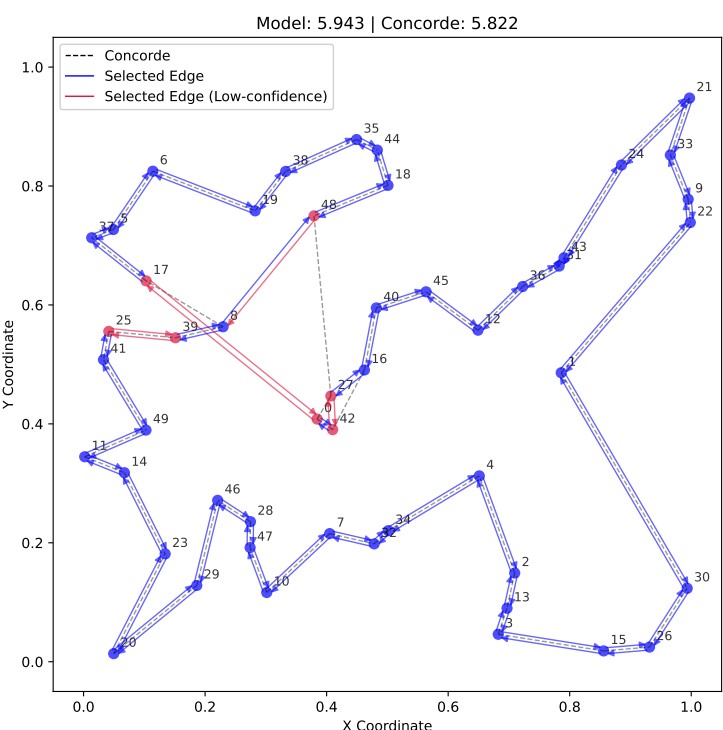

Figure 8: Example of fallback sequential greedy tour construction (GE) using the same heatmap in Figure 7

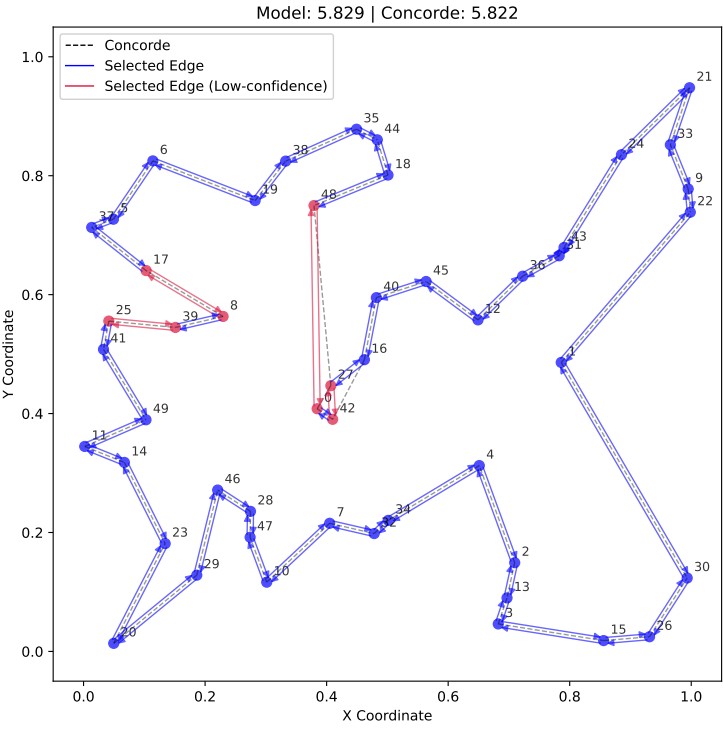

Figure 9: Example of fallback sequential best sampled tour (SE) construction using same heatmap in Figure 7

# E    ACKNOWLEDGEMENT OF LLM USAGE

In this work, large language models (LLMs) were utilized solely for assisting with the drafting and polishing of the manuscript text, including suggestions for phrasing, grammar, and structural refinements. All core scientific contributions, such as the development of the methodology, design of experiments, data analysis, and interpretation of results, were performed independently by the authors without reliance on LLMs for generating ideas, code, or substantive content.

