# OpenReview forum: "Efficiently Solving the TSP in One-Shot Without Post-Processing"
_ICLR.cc/2026/Conference — Submitted to ICLR 2026_

### Official Review · Reviewer_AKcX · 2025-10-29

**Soundness:** 2
**Presentation:** 3
**Contribution:** 2
**Rating:** 2
**Confidence:** 5

**Summary:**

The paper proposes a one-shot, non-autoregressive TSP approach that predicts a dense edge-score matrix and constructs tours by selecting the top-2 incident edges per node in parallel, avoiding search-based post-processing. It trains a single model across sizes via unsupervised self-improvement and introduces NAR-specific metrics (edge precision, tour validity); experiments report near-optimal tours from a single forward pass.

**Strengths:**

The paper’s motivation is clear: it addresses the confound where search/post-processing can dominate neural TSP results and latency, and instead commits to a one-shot, no-post-processing setup. This goal is stated upfront and carried consistently through the method and evaluation, making the claims easy to verify.

The pipeline—predict an edge-score matrix, then decode with a simple per-node rule—is clearly written and easy to follow.

Finally, the reported metrics are aligned with the premise (e.g., edge precision, tour validity), which helps interpret outcomes as model ability rather than search effects.

**Weaknesses:**

1. The empirical results feels partial. Several experiment tables have missing entries; key baselines (Att-GCN, DIMES, DIFUSCO, SoftDist) only appear on TSP-500. This weakens claims about broad superiority.

2. SIL is positioned as label-free, but so are standard RL and unsupervised surrogates. It will be better to make it clear why SIL is chosed over other training methods, such as a small ablation that pits SIL against REINFORCE/rollout and a simple unsupervised baseline, with convergence speed, stability, and final gap, would clarify the trade-offs.

3. All experiments are on uniform Euclidean instances. Generalization remains unclear under distribution shift. It would strengthen the paper to add clustered, anisotropic, and mixed distributions (see the variety used in [1]) and to report cross-distribution transfer.

4. Finally, some design choices (notably the value of K) are justified qualitatively (“learns faster with more diverse batches”) but lack evidence. A brief appendix study sweeping K—with validity rate, gap distributions, and training dynamics—would ground the choice and reveal any speed–quality trade-offs.

[1] Fang et al., “INViT: a generalizable routing problem solver with invariant nested view transformer.”

**Questions:**

1. Could you elaborate on the rationale behind the Sequential Edge Selection(Algorithm 1) design? As presented, it remains a heuristic, and the underlying mechanism is not fully clear. In particular, it would help to explain the guiding principles for the selection order, tie-breaking, and interactions with the predicted scores, and why this specific procedure is preferable to other reasonable choices.

2. Could you more clearly explain the intended meaning of the new metrics (Optimal Edges, Valid Tours, Optimal Tours) and how they relate to final tour length? From the reported results, their values are not in a simple positive/negative correlation relationship with one another, nor with the final performance across different decoding methods. It would be helpful to articulate the expected relationships and the role of the decoder in mediating these metrics and the final tour cost.

---

> ### Author Response · Authors · 2025-11-26
> **Response to Reviewer AKcX (1/3)**
>
> We thank the reviewer for their careful reading and constructive feedback.
>
> **Addressing the Weaknesses:**
> > 1. The empirical results feels partial. Several experiment tables have missing entries; key baselines (Att-GCN, DIMES, DIFUSCO, SoftDist) only appear on TSP-500. This weakens claims about broad superiority.
>
> - Att-GCN results are already reported for all problem sizes (TSP20–500) in the original submission.
>
> - The authors of DIMES did not publish their results or model checkpoints for TSP sizes 20, 50, 100 and 200. Therefore, it is not possible for us to evaluate performance of DIMES on these graph sizes. This should weaken claims about broad superiority of DIMES, not our model. Moreover, on TSP500 (the only size where direct comparison is possible), DIMES is the weakest baseline by a considerable margin across all decoding schemes.
>
> - Missing results for DIFUSCO and SoftDist have been updated. Please see our common response to all reviewers for complete details.
>
> > 2. SIL is positioned as label-free, but so are standard RL and unsupervised surrogates. It will be better to make it clear why SIL is chosed over other training methods, such as a small ablation that pits SIL against REINFORCE/rollout and a simple unsupervised baseline, with convergence speed, stability, and final gap, would clarify the trade-offs.
>
> We tried to train the model using REINFORCE, but we could not train due to instability. The gradient norm and validation cost would explode randomly after some period of training. In contrast, SIL was extremely stable throughout training, with smooth convergence and no such failures.
>
> Self-Improvement Learning (SIL) outperformed Supervised Learning (SL) in our experiments. It may seem counter-intuitive, given that SL is trained on ground-truth optimal labels, whereas SIL relies on noisier pseudo-labels which are suboptimal. The key reason is data scale: SL is constrained to a fixed dataset (100,000 labelled instances in this case), while SIL can generate and learn from a practically unlimited number of unlabelled instances with self-generated targets. This massive increase in effective training data more than compensates for the reduced label quality, leading to superior final performance.
>
> Therefore, SIL was the clear choice to train our model in an unsupervised manner. Please refer to Appendix B of our revised paper for full details.
>
> > 3. All experiments are on uniform Euclidean instances. Generalization remains unclear under distribution shift. It would strengthen the paper to add clustered, anisotropic, and mixed distributions (see the variety used in [1]) and to report cross-distribution transfer.
>
> We have added two sets of experiments in Appendix C of the revised paper:
>
> (a) Cross-distribution generalisation: performance of our original model (trained only on uniform Euclidean instances) on the Clustered, Explosion, and Implosion distributions from [1].
>
> (b) Multi-distribution training: a new model with identical architecture trained from scratch on a mixed dataset containing all four distributions (Uniform, Clustered, Explosion, Implosion).
>
> The results show reasonable generalisation from uniform training and even stronger performance when all distributions are included in training, confirming that our approach is not inherently limited to uniform Euclidean graphs.
>
> Additionally, we have also reported strong generalisation of our model's performance on unseen graph sizes. Please refer to Appendix C for full details.
>
> > 4. Finally, some design choices (notably the value of K) are justified qualitatively (“learns faster with more diverse batches”) but lack evidence. A brief appendix study sweeping K—with validity rate, gap distributions, and training dynamics—would ground the choice and reveal any speed–quality trade-offs.
>
> We have addressed this by conducting the study you suggested and we have now reported these results in Appendix A of our updated paper. The study clearly indicates that training without augmentation ($K=1$) yields the worst performance. In contrast, the best results are achieved with the smallest degree of augmentation ($K=2$). Adding more augmented copies beyond this point degrades performance, most likely because excessive repetition of the same underlying data reduces the effective information density available to the model during training.

---

> ### Author Response · Authors · 2025-11-26
> **Response to Reviewer AKcX (2/3)**
>
> **Answers to the Questions:**
> > 1. Could you elaborate on the rationale behind the Sequential Edge Selection(Algorithm 1) design?
>
> We would like to clarify that Sequential Edge Selection is not a new heuristic introduced in our work; the same procedure was already used in DIFUSCO (though not explained in detail there). We adopted and clearly described it because empirical evidence in Table 1 shows it consistently constructs better tours from the same heatmap compared to traditional node-by-node construction (GE/SE outperform G/S across all baselines).
>
> > As presented, it remains a heuristic, and the underlying mechanism is not fully clear. In particular, it would help to explain the guiding principles for the selection order, tie-breaking, and interactions with the predicted scores,
>
> - The guiding principle for the selection order has been already explained in Algorithm 1 (lines 8 to 11). But let us explain it further to remove any confusion - At each step, among all currently valid candidate edges (those that would not create degree greater than 2 for any node or would not create a premature cycle of length < N), we either:
>     1. greedily pick the highest-probability edge, or
>     1. sample from the probability distribution over valid candidates.
>
> - There is no tie-breaking involved anywhere.
>
> > and why this specific procedure is preferable to other reasonable choices.
>
> - This specific procedure is preferable due to the experimental results we have reported in Table 1 of our paper, which confirm that given the same heatmap, the Sequential Edge construction method constructs more optimal (smaller tour length) TSP tours than the Sequential Node construction method. It can be observed in Table 1 that results for the Sequential Edge construction (denoted by $GE$ for Greedy and $SE$ for Sampled) outperform the corresponding Sequential Node construction of the same baseline (denoted by $G$ for Greedy and $S$ for Sampled) performed on the same heatmaps (model output).
> - We invite the reviewer to specify alternative reasonable choices for further discussion.

---

> ### Author Response · Authors · 2025-11-26
> **Response to Reviewer AKcX (3/3)**
>
> > 2. Could you more clearly explain the intended meaning of the new metrics (Optimal Edges, Valid Tours, Optimal Tours)
>
> The intended meaning of the new metrics are as follows:
>
> - **Optimal Edges**: Percentage of the model’s top-2 predicted edges per node that belong to the optimal tour given by Concorde.
> - **Valid Tours**: Percentage of instances where selecting the top-2 predicted edges per node forms a single valid Hamiltonian cycle.
> - **Optimal Tours**: Percentage of instances where the top-2 predicted edges per node form the exact optimal tour given by Concorde.
>
> > and how they relate to final tour length?
>
> These metrics provide a way to measure model performance when tour length is *undefined* in case of invalid tours. Generally, higher values for these metrics should lead to lower final tour lengths.
>
> In our paper, we want to measure the raw performance of ML models in solving the TSP, without its performance being distorted by post-processing techniques. We do that by taking the top two edges predicted per node by the model, and measuring how close these predictions are to the optimal solution. But, these predictions are not guaranteed to be valid TSP (Hamiltonian cycle) and tour length is undefined for an invalid Hamiltonian cycle. This is why we need these additional metrics to give us insights into raw model performance.
>
> Let us illustrate this point with a simple example for TSP10 (a graph with $N=10$):
>
> - Case A: Out of the 10 optimal edges which would construct the optimal solution, Model A predicts 7 of them accurately. The rest 3 edges predicted by the model are suboptimal. But the overall selection still forms a valid hamiltonian cycle.
> - Case B: Same as Case A, Model B predicts 7 optimal edges while 3 are suboptimal, but the overall selection is *not* a valid hamiltonian cycle.
> - Case C: Model C predicts 2 out of the 10 optimal edges, failing to form a valid hamiltonian cycle.
>
> We can calculate the tour length only for Case A. So how can we compare it with Case B and C? This is where these metrics come into play.
>
> By looking at the **Optimal Edges** metric, we can see that A (7) = B (7) > C (2). So, clearly A and B are better than C. The **Valid Tours** metric shows us that Model A manages to predict a valid tour while Model B and C fail. So, A is better than B.
>
> The **Optimal Tours** metric indicates that none of the models manage to predict the optimal tour, which means the models haven't yet reached optimality.
>
> > From the reported results, their values are not in a simple positive/negative correlation relationship with one another, nor with the final performance across different decoding methods. It would be helpful to articulate the expected relationships and the role of the decoder in mediating these metrics and the final tour cost.
>
> - Higher values on all three metrics indicate better raw predictions.
> - They are positively correlated with each other and inversely correlated with final tour length when a valid tour exists.
> - Mathematically, since Optimal Tours are a subset of Valid Tours, this relationship will always hold: Optimal Tours ≤ Valid Tours.
> - From the results in Table 2 in our paper, there is a clear pattern: Optimal Edges (easiest) > Valid Tours > Optimal Tours (hardest).
> - It is easy to see that the Optimal Edges is the easiest metric, since it is possible to get close to 70% optimal edges just by predicting the two shortest edges, as demonstrated by SoftDist. But doing so would not lead to even a single valid TSP tours, as can be seen in *Valid Tours* score of SoftDist.

---

> > ### Comment · Reviewer_AKcX · 2025-11-27
> >
> > Thank you for the detailed response and the added experiments (distribution shift, K-sweep, and SIL vs RL/SL).
> >
> > I still have remaining questions related to my original Question 2 on the new metrics:
> >
> > - On TSP-500, UTSP’s Optimal Edges are lower than Att-GCN’s, yet its tour length is shorter.
> > - In the common response, DIFUSCO×50 has higher Optimal Edges and Valid Tours than your unified model, but still yields worse final tour length.
> >
> > What is the main reason for this mismatch between the metrics and the final cost?
> > Is it primarily due to the sequential edge selection heuristic (GE/SE), or are there other systematic effects?

---

### Official Review · Reviewer_X36X · 2025-10-30

**Soundness:** 2
**Presentation:** 3
**Contribution:** 2
**Rating:** 2
**Confidence:** 4

**Summary:**

The paper proposes a new neural non-autoregressive approach for solving TSPs. Specifically, the proposed approach is utilizing data augmentation and self-improvement learning scheme to train the model without requiring expert solutions. The emphasis of the paper is on one-shot generation without utilizing post-processing techniques such as beam search, 2-opt, MCTS, etc. Experimental results show the model outperforms neural baselines on TSP problems of sizes ranging from 20 to 500.

**Strengths:**

- Novel neural NAR approach for TSP, in particular it proposes a new data augmentation and self-improvement training scheme.
- The experiments show the proposed approach outperforms the baselines in terms of performance gap.

**Weaknesses:**

- Limited technical novelty: despite improved performance, the paper makes a limited technical contributions. It seems the main innovation behind the proposed approach is the self-improvement procedure and the data augmentation.
- The experimental results have several major concerns:
	* As the paper did not run the baselines and only used results reported in previous papers, the main table of results is missing 80% of the results for three of the baselines including the strongest baseline of DIFUSCO. It is therefore not clear that the proposed model is the best across all problem sizes.
	* When focusing on the last column (TSP500) which is the only one with full results, the observed gains in standard greedy search compared to DIFUSCO are limited, and when considering the best setting for the proposed approach (SIL+SE), both LEHD and BQ-NCO are outperforming the proposed approach with LEHD seems faster (not clear why it is written as 0.3m instead in seconds other approaches) and BQ-NCO a bit slower. In the only setting considered that includes results for DIFUSCO (TSP500), it also outperforms the proposed approach in terms of # of optimal edges and valid tours
	* Results for DIFUSCO for both greedy and sampling are identical however in the original paper sampling performed better on TSP-500
	* Results are restricted to TSP and do not consider other popular problems such as CVRP, MIS, OP, etc. It is not clear how the observed patterns generalize beyond TSP.
	* No analysis on out-of-distribution performance (either sizes that were not considered in training or different data distributions considered in previous works).
	* It is not clear why SIL+SE is ~20 times slower than SIL+GE - what is the source of this slow down?
- This is not a critical point, but I am not sure I am convinced that it makes sense to exclude simple heuristics like 2-opt, while including constraints on the degree and merging connected components as part of the decoding.

**Questions:**

See my questions and concerns under weaknesses above

---

> ### Author Response · Authors · 2025-11-26
> **Response to Reviewer X36X (1/3)**
>
> We thank the reviewer for their careful reading and constructive feedback.
>
> **Addressing the Weaknesses/Questions:**
> > - Limited technical novelty: despite improved performance, the paper makes a limited technical contributions. It seems the main innovation behind the proposed approach is the self-improvement procedure and the data augmentation.
>
> We would like to clarify the core technical contribution of our work. While the self-improvement procedure and data augmentation certainly play an important role in achieving strong empirical performance, we believe the primary novelty lies elsewhere: Our core contribution is demonstrating, for the first time to the best of our knowledge, that it is *possible* for a neural network to directly output a valid (and often near-optimal) TSP tour in a single forward pass, without any form of sequential tour construction, autoregressive decoding, or post-processing methods. No prior work has successfully pursued this exact one-shot interpretation of model outputs as complete tours.
>
> We achieved valid TSP tours on roughly 70% of the dataset (about 7,000 out of 10,000 problems) for the non-trivial graph size of $N=100$, and even better results on smaller sizes. While we acknowledge that we could not achieve valid TSP solutions for every single graph in the test dataset (i.e. 100% *Valid Tours*), we believe these results are still sufficient to strongly validate the proof of concept and open a promising research direction, especially considering that we achieved these results on moderate computational power ($4\times$ RTX 2080 Ti). With greater compute, longer training, larger models, or architectural improvements in future research, we expect substantial further gains.
>
> > - As the paper did not run the baselines and only used results reported in previous papers, the main table of results is missing 80% of the results for three of the baselines including the strongest baseline of DIFUSCO. It is therefore not clear that the proposed model is the best across all problem sizes.
> > - When focusing on the last column (TSP500) which is the only one with full results, the observed gains in standard greedy search compared to DIFUSCO are limited,
> > ...
> > In the only setting considered that includes results for DIFUSCO (TSP500), it also outperforms the proposed approach in terms of # of optimal edges and valid tours
>
> - Please see our detailed common response to all reviewers where we have given details about most of the missing results that we have now added in our updated manuscript.
>
> - The comparison with DIFUSCO, as reported in the original submission had several important caveats that were not sufficiently highlighted. A detailed and updated comparison is provided in our common response.
>
> > and when considering the best setting for the proposed approach (SIL+SE), both LEHD and BQ-NCO are outperforming the proposed approach with LEHD seems faster and BQ-NCO a bit slower.
>
> We would like to highlight that *efficiency* is a key focus of our work and so while it is true that the two AR approaches indeed perform slightly better than our approach, they are still less efficient. These AR methods require $N$ forward passes of the model for problems of size $N$. So, for TSP500, they require **500** forward passes, while our approach requires just **1** forward pass. That is a $500\times$ difference in efficiency, and we still achieve competitive results against these methods.
>
> Moreover, it is also important to highlight that these two AR approaches are supervised methods trained on optimal ground-truth solutions, whereas our approach is fully unsupervised and does not require access to optimal tours during training.

---

> ### Author Response · Authors · 2025-11-26
> **Response to Reviewer X36X (2/3)**
>
> > - Results for DIFUSCO for both greedy and sampling are identical however in the original paper sampling performed better on TSP-500
>
> The discrepancy arises because the authors of DIFUSCO used a different definition of *sampling* decoding scheme than ours. Quoting the DIFUSCO paper (page 8):
>
> > In general, we find that 50 (diffusion steps) × 1 (samples) policy and 10 (diffusion steps) × 16 (samples) policy make a good balance between exploration and exploitation for discrete DIFUSCO models and use them as the Greedy and Sampling strategies for the rest of the experiments.
>
> This means, for the sampling results in the DIFUSCO paper, they generated 16 separate heatmaps for each problem, and then performed a greedy selection on each of the heatmaps (same as our GE decoding), and reported the best result. On the other hand, for each baseline (including DIFUSCO) in our paper, we used a single heatmap per problem to perform all the decoding schemes (G, S, GE, SE). The S/SE decoding schemes we used takes the best trajectory produced from 16 sampled + 1 greedy from the same heatmap.
>
> Furthermore, the reason for identical greedy (G) and sampling (S) results for DIFUSCO in our paper is that for every instance in the test dataset, the greedy tour happened to be the best among the 17 tours (1 greedy + 16 $\times$ sampled) generated for each of the problem instances. (S/SE reports the best among the greedy + sampled trajectories).
>
> > - Results are restricted to TSP and do not consider other popular problems such as CVRP, MIS, OP, etc. It is not clear how the observed patterns generalize beyond TSP.
>
> We fully acknowledge that our current experiments are limited to TSP. However, TSP remains a foundational benchmark in Neural Combinatorial Optimization (NCO) and Computer Science in general, as evident from lots of papers being published on it in recent years in top ML venues. While we do plan to extend our work to other NCO problems and routing problems in a future work, in our view, this work demonstrating a new paradigm - one-shot, efficient post-processing-free TSP solution - on this important problem, is a meaningful standalone contribution.
>
> > - No analysis on out-of-distribution performance (either sizes that were not considered in training or different data distributions considered in previous works).
>
> We have updated the paper with analysis on both unseen sizes and different data distributions. Please refer to Appendix C in our revised paper.
>
> For different data distributions, we have added two sets of experiments:
>
> (a) Cross-distribution generalisation: performance of our original model (trained only on uniform Euclidean instances) on the Clustered, Explosion, and Implosion distributions from [1].
>
> (b) Multi-distribution training: a new model with identical architecture trained from scratch on a mixed dataset containing all four distributions (Uniform, Clustered, Explosion, Implosion).
>
> The results show reasonable generalisation from uniform training and even stronger performance when all distributions are included in training, confirming that our approach is not inherently limited to uniform Euclidean graphs.
>
> For evaluation on problem sizes not seen during training, we evaluated our model on $N \in \{15, 30, 75, 150, 300, 600\}$. The results show excellent generalisation of our model across unseen sizes. Please refer to Appendix C for full details.

---

> ### Author Response · Authors · 2025-11-26
> **Response to Reviewer X36X (3/3)**
>
> > - It is not clear why SIL+SE is ~20 times slower than SIL+GE - what is the source of this slow down?
>
> Since SE generates 17 trajectories (1 greedy + 16 sampled) compared to just the 1 greedy trajectory for GE, it is expected to be slower. If we generated the 17 trajectories one after the other, ~17 times slower for about ~17 times more work is what we would expect. But our PyTorch implementation generates the trajectories in parallel. So, it is indeed surprising that it is ~17 times slower. While we would certainly investigate the reason for this, we believe this is an implementation detail and should not affect the core ideas of our work.
>
> > - This is not a critical point, but I am not sure I am convinced that it makes sense to exclude simple heuristics like 2-opt, while including constraints on the degree and merging connected components as part of the decoding.
>
> We excluded 2-Opt results in the original submission because it is a post-processing step applied after a complete tour exists. In contrast, the degree constraints and component merging in sequential decoding (GE/SE) are validity-enforcement mechanisms applied during construction from an incomplete subgraph.
>
> That said, we agree that 2-Opt results can be informative. For completeness, here are the results after applying 2-Opt to our originally reported results:
>
> | Decoding Scheme | TSP20         | TSP50         | TSP100        | TSP200        | TSP500        |
> |----------------:|--------------:|--------------:|--------------:|--------------:|--------------:|
> | G + 2-Opt       | 3.831 (-0.001%) | 5.693 (0.022%) | 7.773 (0.108%) | 10.757 (0.272%) | 16.807 (1.350%) |
> | S + 2-Opt       | 3.830 (-0.005%) | 5.692 (0.009%) | 7.772 (0.095%) | 10.763 (0.322%) | 16.810 (1.365%) |
> | GE + 2-Opt      | 3.830 (-0.003%) | 5.692 (0.006%) | 7.769 (0.060%) | 10.751 (0.212%) | 16.785 (1.214%) |
> | SE + 2-Opt      | 3.830 (-0.004%) | 5.692 (0.000%) | 7.768 (0.042%) | 10.746 (0.164%) | 16.779 (1.179%) |

---

### Official Review · Reviewer_mSs4 · 2025-10-31

**Soundness:** 2
**Presentation:** 2
**Contribution:** 2
**Rating:** 4
**Confidence:** 3

**Summary:**

This paper proposes a non-autoregressive (NAR) neural approach to solving the Traveling Salesman Problem (TSP) in a single forward pass without relying on post-processing techniques such as beam search, MCTS, or 2-opt. The model outputs an edge probability heatmap from which the final tour is constructed by selecting the top two edges per node. The method is trained using Self-Improvement Learning (SIL), an unsupervised framework where the model iteratively improves through self-generated pseudo-labels, avoiding the need for expert solutions or reinforcement learning. It also introduces new evaluation metrics to assess the model’s raw predictive capability without post-processing. Experiments on standard TSP benchmarks show that the approach achieves near-optimal results on smaller graphs and competitive results on larger instances, while being significantly faster.

**Strengths:**

1. The one-shot, non-autoregressive design drastically reduces inference time compared to AR or diffusion-based methods.

2. Achieves near-optimal tours on small to medium graphs; valid and optimal tours are significantly higher than prior NAR methods.

**Weaknesses:**

1. Performance drops notably for large TSPs (e.g., 500 nodes), suggesting limitations in model capacity or generalization.
2. While the approach is efficient, its novelty primarily lies in combining existing ideas rather than introducing a fundamentally new model.
3. This paper claims about outperforming others “without post-processing” could be better qualified, as some baselines use additional refinement by design.

**Questions:**

1. How does SIL compare quantitatively to reinforcement learning or supervised approaches in terms of convergence speed and stability?
2. Could enforcing global connectivity constraints during decoding further improve valid tour rates?

---

> ### Author Response · Authors · 2025-11-26
> **Response to Reviewer mSs4 (1/2)**
>
> We thank the reviewer for their careful reading and constructive feedback.
>
> **Addressing the Weaknesses:**
> > 1. Performance drops notably for large TSPs (e.g., 500 nodes), suggesting limitations in model capacity or generalization.
>
> A performance degradation when scaling from small/medium to large instances is expected in almost all learning-based TSP solvers, because problem difficulty grows quadratically with the number of nodes while model capacity is fixed. The same model that solves TSP100 almost perfectly will naturally struggle more on TSP500. This is not a sign of unusually poor generalisation but a standard scaling behavior observed across the literature (e.g., AM, POMO, DIFUSCO etc.). We believe our results remain competitive with or superior to existing end-to-end trainable methods of comparable capacity.
>
> > 2. While the approach is efficient, its novelty primarily lies in combining existing ideas rather than introducing a fundamentally new model.
>
> We respectfully note that most impactful advances in deep learning for combinatorial optimization arise from thoughtful integration and refinement of existing components rather than entirely new architectural paradigms. Very few papers in this area introduce a truly "fundamentally new" architectural paradigm.
>
> Our core contribution is demonstrating, for the first time to the best of our knowledge, that it is *possible* for a neural network to directly output a valid (and often near-optimal) TSP tour in a single forward pass, without any form of sequential tour construction, autoregressive decoding, or post-processing methods. No prior work has successfully pursued this exact one-shot interpretation of model outputs as complete tours.
>
> We achieved valid TSP tours on roughly 70% of the dataset (about 7,000 out of 10,000 problems) for the non-trivial graph size of $N=100$, and even better results on smaller sizes. While we acknowledge that we could not achieve valid TSP solutions for every single graph in the test dataset (i.e. 100% *Valid Tours*), we believe these results are still sufficient to strongly validate the proof of concept and open a promising research direction, especially considering that we achieved these results on moderate computational power ($4\times$ RTX 2080 Ti). With greater compute, longer training, larger models, or architectural improvements in future research, we expect substantial further gains.
>
>
> > 3. This paper claims about outperforming others “without post-processing” could be better qualified, as some baselines use additional refinement by design.
>
> Using additional refinement comes at a computational cost. Therefore, achieving comparable or better performance *without* any refinement is a meaningful practical advantage: significantly faster inference and lower deployment cost. Nothing prevents us from applying additional refinement on top of our approach, if desired. So, the ability to match refine-augmented baselines without it is a strength, not a weakness.
>
> In fact, in response to reviewer X36X, we have now additionally reported the results of our model after applying the 2-Opt refinement, where our approach shows excellent performance. This shows that our proposed solution remains fully compatible with additional post-processing. These results are reported in our common response to all reviewers.
>
> We are happy to further clarify the text to avoid any ambiguity.
>
> **Answers to the Questions:**
> > 1. How does SIL compare quantitatively to reinforcement learning or supervised approaches in terms of convergence speed and stability?
>
> We tried to train the model using RL, but we could not train due to instability. The gradient norm and validation cost would explode randomly after some period of training. In contrast, SIL was extremely stable throughout training, with smooth convergence and no such failures.
>
> Self-Improvement Learning (SIL) outperformed Supervised Learning (SL) in our experiments. It may seem counter-intuitive, given that SL is trained on ground-truth optimal labels, whereas SIL relies on noisier pseudo-labels which are suboptimal. The key reason is data scale: SL is constrained to a fixed dataset (100,000 labelled instances in this case), while SIL can generate and learn from a practically unlimited number of unlabelled instances with self-generated targets. This massive increase in effective training data more than compensates for the reduced label quality, leading to superior final performance.
>
> Therefore, SIL was the clear choice to train our model in an unsupervised manner. Please refer to Appendix B of our revised paper for full details.

---

> ### Author Response · Authors · 2025-11-26
> **Response to Reviewer mSs4 (2/2)**
>
> > 2. Could enforcing global connectivity constraints during decoding further improve valid tour rates?
>
> The most natural way to enforce global connectivity constraints in our Transformer architecture is via attention masking (e.g., k-nearest-neighbor masking). However, it is not feasible to apply the attention mask only during decoding. Any mask applied at test time must also be used during training (and vice-versa). During experimentation we found that k-NN masking provides an initial boost but both masked and unmasked models ultimately converge to nearly identical performance. Because computing the k-NN graph incurs non-negligible overhead for each problem instance during both training and testing, while yielding no final gain, we opted for the simpler unmasked model. We are happy to include these results in the next revision if requested.

---

### Official Review · Reviewer_tPHC · 2025-11-01

**Soundness:** 3
**Presentation:** 4
**Contribution:** 2
**Rating:** 4
**Confidence:** 3

**Summary:**

This paper presents a non-autoregressive (NAR) approach for solving TSP in one-shot without post-processing. The key innovation is a two-selection scheme that extracts valid tours from heatmaps by selecting top two edges per node. Training uses Self-Improvement Learning, an unsupervised method with self-generated pseudo-labels. On TSP-20, it achieves 98.23% valid tours and 0.011% gap, outperforming Att-GCN. A unified model handles N=20 to N=500.

**Strengths:**

The two-selection scheme achieves 98.23% valid tours on TSP-20 versus Att-GCN's 72.49%, with mathematical derivation ensuring two edges per node. SIL training eliminates expert solution dependence while achieving near-optimal performance. Three new metrics—Optimal Edges, Valid Tours, Optimal Tours—provide fairer NAR assessment; prior methods achieve 0% valid tours while this maintains 70.24% on TSP-100. Unified cross-scale model handles N=20-500.

**Weaknesses:**

Performance degrades severely on large instances: TSP-500 shows only 0.78% valid tours and 6.034% gap versus LEHD's 1.560%, with no N≥1000 evaluation. Valid tour percentage drops dramatically, meaning 99.22% of TSP-500 instances fail. The paper lacks failure mode analysis and recovery mechanisms. While claiming "post-processing-free," tour completion remains necessary for large-scale deployment. Comparisons focus on NAR methods; broader AR comparisons beyond POMO would strengthen evaluation.

**Questions:**

Can the method scale to N≥1000, and what prevents evaluation at this scale?
What specific failure modes cause the dramatic valid tour drop on large instances?

---

> ### Author Response · Authors · 2025-11-26
> **Response to Reviewer tPHC**
>
> We thank the reviewer for their careful reading and constructive feedback.
>
> **Addressing Weaknesses:**
> > Performance degrades severely on large instances: TSP-500 shows only 0.78% valid tours and 6.034% gap versus LEHD's 1.560%
>
> While we acknowledge the larger optimality gap of our method (6.034%) compared to LEHD (1.560%) on TSP500, it is important to emphasize that LEHD is a supervised method trained on optimal ground-truth solutions, whereas our approach is fully unsupervised and does not require access to optimal tours during training.
>
> > The paper lacks failure mode analysis and recovery mechanisms.
>
> In the context of single-step (one-shot) tour generation, traditional "recovery mechanisms" are not applicable, as backtracking even a single step would return the process to the initial state. Nevertheless, we have reported our results using several fallback sequential construction methods (G, S, GE, SE) in Table 1, which serve as practical recovery strategies when the one-shot approach produces an invalid tour.
>
> We have also added examples of both success and failure cases, including fallback solutions in case of failure, in Appendix D.
>
> > While claiming "post-processing-free," tour completion remains necessary for large-scale deployment.
>
> We thank the reviewer for raising this point and would like to clarify the terminology. By "post-processing", we refer to methods like 2-Opt, 3-Opt etc. that take a complete valid tour as input and iteratively improve it. Simple sequential tour construction with greedy or sampling-based edge selection without backtracking is not considered post-processing in this context.
>
> Our primary contribution is a direct one-shot generation approach that requires neither post-processing nor sequential construction. Although this method does not yet achieve 100% validity on entire test datasets especially on larger problem sizes, the fallback sequential construction mechanisms (G, S, GE, SE) still produce valid, near-optimal tours without any advanced post-processing. Thus, our claim of being "post-processing-free" remains accurate, as we do not use post-processing methods like MCTS or 2-opt/k-opt.
>
> > Comparisons focus on NAR methods; broader AR comparisons beyond POMO would strengthen evaluation.
>
> As our method is NAR, we naturally focused the main comparison on other NAR baselines. However, we already included comparisons with strong AR methods beyond POMO - namely LEHD and BQ-NCO.
>
> **Answers to Questions:**
> > Can the method scale to N≥1000, and what prevents evaluation at this scale?
>
> There is no fundamental limitation in our approach that prevents scaling to N ≥ 1000. The only constraint was computational resources: training a model to full convergence at this scale was not feasible within our budget. Reporting results from an under-trained model would be misleading when compared to prior work that trained properly at these scales. We therefore chose not to include such results.
>
> > What specific failure modes cause the dramatic valid tour drop on large instances?
>
> The reason is that even a single incorrect edge prediction would make the entire tour invalid. This can be easily understood with an illustrative example. Let's say for both TSP20 and TSP500, a model makes 500 edge predictions, out of which 1 is incorrect, then:
>
>
> ||TSP20|TSP500|
> |--:|--:|--:|
> | Problem instances | 25 | 1 |
> | Total predictions made by model | 25×20 = 500 | 1×500 = 500 |
> | Total prediction errors | 1/500 | 1/500 |
> | Valid Tours | 24/25 = 96% | 0/1 = 0% |
>
> Thus, even with the same edge prediction error rate, tour validity collapses dramatically on larger instances. This is an inherent challenge of one-shot tour generation and explains the observed behavior.

---

### Author Response · Authors · 2025-11-26
**Common Response Addressed to All Reviewers (1/3)**

We thank the reviewers for their thoughtful and constructive feedback, which has helped us strengthen the manuscript in several important ways.

## List of changes in updated paper

- Missing results for DIFUSCO and SoftDist have been updated (see details below)
- There was an error in the original submission where we mistakenly reported the Optimality Gap for greedy decoding (G) of DIMES as 89.325% (that is in fact the value for Att-GCN right above it). The correct value of 210.219% has now been updated.
- A brief study showcasing the choice of data augmentation factor K has been added to Appendix&nbsp;A.
- A comparison of Self-Improvement Learning (SIL) with Reinforcement Learning (RL) and Supervised Learning (SL) been added to Appendix&nbsp;B.
- Out of distribution evaluation on unseen problem sizes as well as different data distributions have been added to Appendix&nbsp;C.
- Illustrative examples of success and failure cases are added in Appendix&nbsp;D.


## Missing results for DIFUSCO and SoftDist have been updated

- We acknowledge the omission of DIFUSCO results on TSP50 and TSP100 (where the authors did provide code, results and model checkpoints). We used the model checkpoints to generate the heatmaps (both for 1 denoising step, and 50 denoising steps) and ran our various decoding strategies to generate the results. We have now added these missing entries in the revised manuscript. Please note that the DIFUSCO results for TSP20 and TSP200 still remain missing as the authors neither reported results, nor published model checkpoints for these sizes. We hope this will not penalize our submission, as the missing data is outside our control.

- The SoftDist authors did not report results for TSP20, TSP50, TSP100, or TSP200. But using their publicly available grid-search code, we identified suitable temperature parameters for each size:

    | Graph Size  | Temperature|
    | -----:      | ----------:|
    | 20          | 0.0403     |
    | 50          | 0.106      |
    | 100         | 0.017      |
    | 200         | 0.013      |

    With these temperatures, we generated heatmaps and evaluated all decoding strategies
    across all the problem sizes. We have updated our paper with these results.

## Missing data for DIMES

- The authors of DIMES did not publish their results or model checkpoints for TSP sizes 20, 50, 100 and 200. Therefore, it is not possible for us to evaluate performance of DIMES on these graph sizes. However, on TSP500 (the only size where direct comparison is possible), DIMES is the weakest baseline by a considerable margin across all decoding schemes.

## Comparison with DIFUSCO
There are a few important points to keep in mind when comparing our results with DIFUSCO:

1. DIFUSCO is a diffusion model which uses multiple denoising steps (forward passes through the model) for generating heatmaps. The reported results for DIFUSCO in the paper were based on heatmaps generated through **50** forward passes of the model! On the other hand, our model generates heatmaps in a single pass. Considering the main focus of our work is to *efficiently* solve the TSP, this key difference can not be ignored. We admit that we did not sufficiently highlight this matter in our paper.

    To rectify this, for a more fair comparison with our work, we have updated the paper with results of DIFUSCO for both 50 passes ($\times 50$), as well as a single pass ($\times 1$) through the diffusion model. It can be clearly seen that, the results of DIFUSCO are significantly degraded when only one pass through the model ($\times 1$) is used to generate the heatmaps. The performance gap between our approach and DIFUSCO $\times 1$ is much wider.

1. Separate DIFUSCO models were trained for each problem size, whereas our single unified model was trained on problems sizes ranging from TSP20 to TSP500 which can handle all these problem sizes. The results reported for DIFUSCO comes from a DIFUSCO model specialized on TSP500. For a more fair comparison, we report the results after fine-tuning our unified model on TSP500 below.

1. DIFUSCO was trained through supervised learning which requires expert solutions, whereas our approach is entirely unsupervised.
1. DIFUSCO was trained on more powerful hardware than our model. The authors of DIFUSCO reported (on page 22 of the DIFUSCO paper) that their models were trained on $8\times$ NVIDIA Tesla V100 Volta GPUs, whereas we trained our model using $4\times$ NVIDIA RTX 2080 Ti.

Despite these advantages DIFUSCO had over our model, we still managed to outperform both variants - DIFUSCO $\times 1$ and DIFUSCO $\times 50$, across TSP50, TSP100 and TSP500, in almost every metric. The only exceptions are *Optimal Edges* and *Optimal Tours* in TSP500. Even that advantage goes away when we fairly compare aginst our model which is finetuned on TSP500 as detailed below.

---

> ### Author Response · Authors · 2025-11-26
> **Common Response Addressed to All Reviewers (2/3)**
>
> ## Comparison of our fine-tuned model with DIFUSCO
>
> | Method               | TSP50 Optimal Edges| TSP50 Valid Tours | TSP50 Optimal Tours | TSP100 Optimal Edges | TSP100 Valid Tours | TSP100 Optimal Tours | TSP500 Optimal Edges | TSP500 Valid Tours | TSP500 Optimal Tours |
> |----------------------|--------------------|-------------------|---------------------|----------------------|--------------------|----------------------|----------------------|--------------------|----------------------|
> | DIFUSCO (×1)         | 94.978%            | 42.750%           | 29.660%             | 91.406%              | 3.460%             | 1.270%               | **86.739%**          | 0%                 | 0%                   |
> | DIFUSCO (×50)        | 94.283%            | 88.600%           | 40.340%             | 89.369%              | 41.880%            | 3.610%               | 83.595%              | 2.344%             | 0%                   |
> | Ours (fine-tuned)    | —                  | —                 | —                   | —                    | —                  | —                    | 81.595%              | **3.125%**         | 0%                   |
> | Ours (unified)       | **96.199%**        | **90.100%**       | **53.730%**         | **93.328%**          | **70.240%**        | **15.420%**          | 80.038%              | 0.781%             | 0%                   |```
>
>
> | Method               | Sequential Decoding | TSP50 Cost | TSP50 Gap | TSP100 Cost | TSP100 Gap | TSP500 Cost | TSP500 Gap |
> |---------------------:|-------:|-----------:|----------:|------------:|-----------:|------------:|-----------:|
> | DIFUSCO (×1)         | G      | 6.089      | 6.924%    | 9.348       | 20.349%    | 24.459      | 47.484%    |
> | DIFUSCO (×50)        | G      | 5.768      | 1.334%    | 8.414       | 8.350%     | 18.800      | 13.366%    |
> | Ours (fine-tuned)    | G      | —          | —         | —           | —          | **18.263**  | **10.141%**|
> | Ours (unified)       | G      | **5.743**  | **0.892%**| **7.975**   | **2.710%** | 18.790      | 13.306%    |
> | DIFUSCO (×1)         | S      | 5.798      | 1.843%    | 8.902       | 14.609%    | 24.459      | 47.484%    |
> | DIFUSCO (×50)        | S      | 5.701      | 0.164%    | 7.980       | 2.769%     | 18.800      | 13.366%    |
> | Ours (fine-tuned)    | S      | —          | —         | —           | —          | **18.190**  | **9.702%** |
> | Ours (unified)       | S      | **5.697**  | **0.095%**| **7.826**   | **0.783%** | 18.737      | 12.983%    |
> | DIFUSCO (×1)         | GE     | 5.789      | 1.683%    | 8.314       | 7.048%     | 20.139      | 21.429%    |
> | DIFUSCO (×50)        | GE     | 5.708      | 0.284%    | 8.035       | 3.479%     | 18.283      | 10.244%    |
> | Ours (fine-tuned)    | GE     | —          | —         | —           | —          | **17.146**  | **3.389%** |
> | Ours (unified)       | GE     | **5.698**  | **0.010%**| **7.797**   | **0.415%** | 17.585      | 6.034%     |
> | DIFUSCO (×1)         | SE     | 5.707      | 0.259%    | 7.974       | 2.687%     | 19.882      | 19.887%    |
> | DIFUSCO (×50)        | SE     | 5.700      | 0.146%    | 7.905       | 1.800%     | 17.777      | 7.195%     |
> | Ours (fine-tuned)    | SE     | —          | —         | —           | —          | **16.850**  | **1.604%** |
> | Ours (unified)       | SE     | **5.692**  | **0.010%**| **7.772**   | **0.095%** | 17.048      | 2.797%     |
>
> Now, we can see that our fine-tuned model on TSP500 achieves better *Valid Tours* than both DIFFUSCO $\times 50$ and DIFFUSCO $\times 1$. The DIFFUSCO $\times 1$ variant does get higher *Optimal Edges* (86.739%) than ours (81.595%), due to DIFFUSCO being directly trained on Concorde's optimal solutions. This is why it inherently predicts closer to Concorde's solution. This can also be observed from the fact that additional denoising steps leads to lowering of Optimal Edges (83.595% for DIFUSCO $\times 50$) as the model attempts to form valid tours for the given instance by slightly moving away from some optimal edges.
>
> Now, we can see that our fine-tuned model on TSP500 achieves better *Valid Tours* than both DIFUSCO $\times 50$ and DIFUSCO $\times 1$. The DIFUSCO $\times 1$ variant does get higher *Optimal Edges* (86.739%) than ours (81.595%), due to DIFUSCO being directly trained on Concorde's optimal solutions. This is why it inherently predicts closer to Concorde's solution. This can also be observed from the fact that additional denoising steps leads to lowering of Optimal Edges (83.595% for DIFUSCO $\times 50$), as the model attempts to form valid tours for the given instance by slightly moving away from some optimal edges.

---

> > ### Author Response · Authors · 2025-11-26
> > **Common Response Addressed to All Reviewers (3/3)**
> >
> > ## Results after applying 2-Opt
> >
> > We excluded 2-Opt results in the original submission because it is a post-processing step applied after a complete tour exists. In contrast, the degree constraints and component merging in sequential decoding (GE/SE) are validity-enforcement mechanisms applied during construction from an incomplete subgraph.
> >
> > That said, we agree with reviewer *X36X* that 2-Opt results can be informative. For completeness, here are the results after applying 2-Opt to our originally reported results:
> >
> > | Decoding Scheme | TSP20         | TSP50         | TSP100        | TSP200        | TSP500        |
> > |----------------:|--------------:|--------------:|--------------:|--------------:|--------------:|
> > | G + 2-Opt       | 3.831 (-0.001%) | 5.693 (0.022%) | 7.773 (0.108%) | 10.757 (0.272%) | 16.807 (1.350%) |
> > | S + 2-Opt       | 3.830 (-0.005%) | 5.692 (0.009%) | 7.772 (0.095%) | 10.763 (0.322%) | 16.810 (1.365%) |
> > | GE + 2-Opt      | 3.830 (-0.003%) | 5.692 (0.006%) | 7.769 (0.060%) | 10.751 (0.212%) | 16.785 (1.214%) |
> > | SE + 2-Opt      | 3.830 (-0.004%) | 5.692 (0.000%) | 7.768 (0.042%) | 10.746 (0.164%) | 16.779 (1.179%) |

---

### Meta-Review · Area_Chair_dprF · 2025-12-30

**Summary:**

This work proposes a neural non-autoregressive (NAR) method for solving the traveling salesman problem (TSP) in a one-shot manner (without any sophisticated post-processing). The key steps are: 1) training a powerful attention-based model to generate a high-quality heatmap via self-improvement learning (SIL), 2) selecting the top two edges per node based on the heatmap, and 3) constructing a valid solution with these selected edges. Experimental results show that the proposed method can achieve good performance on TSP instances with up to 500 nodes.

The reviewers have consistently given negative scores (4, 4, 2, 2) on this work and raised many concerns, including limited technical novelty, performance on large-scale instances, valid tour percentage, the claim of being "post-processing-free", the training method, the experimental results, the need for more comparisons, and insufficient analysis. After the rebuttal, Reviewer AKcX raised some follow-up questions, but the other reviewers did not respond. No reviewer indicated that they would increase their score. After reading the paper in detail, I believe the concerns raised by the reviewers are reasonable and many of them have not yet been adequately addressed in the rebuttal.

Therefore, I recommend rejecting this work.

**Reviewer Concerns:**

In the rebuttal, the authors have provided new results for DIFUSCO and SoftDist, a comparison of self-improvement learning (SIL) with reinforcement learning (RL) and supervised learning (SL), an out-of-distribution evaluation, an analysis of the data augmentation, and illustrative examples of success and failure cases. Therefore, I believe some concerns regarding experimental results, more comparisons, and insufficient analysis have been partially addressed.

However, a more in-depth comparison and analysis is still needed. For example, different SIL-based methods have already been proposed for solving combinatorial optimization problems, such as [1,2,3], which are already cited in this work. It is questionable why none of them are compared in this work, especially since [2] and [3] can be directly used to solve TSP instances. In addition, the proposed method is only tested on TSP with up to 500 nodes, whereas other SIL methods like [3] can already solve TSP and CVRP with up to 100k nodes. Therefore, the reviewers' concerns about large-scale instances remain outstanding. Furthermore, extension to other combinatorial optimization problems (e.g., CVRP and many variants, MIS, OP) is highly desirable to support the generalization ability of the proposed method.

Although the rebuttal has also provided some discussion to address other concerns, such as limited technical novelty, valid tour percentage, and the claim of being "post-processing-free", a careful and in-depth revision is still necessary before this work can be accepted by top conferences such as ICLR.

[1] Self-labeling the job shop scheduling problem. NeurIPS 2024.

[2] Self-improvement for neural combinatorial optimization: Sample without replacement, but improvement. TMLR 2024.

[3] Boosting neural combinatorial optimization for large-scale vehicle routing problems. ICLR 2025.

**Reviewer Scores:**

Based on the comments above, I believe that all reviewers would have maintained their original scores if they had been able to participate fully in the discussion.

---

### Decision · Program_Chairs · 2026-01-26

Reject